# Ethnic minority women's empowerment in agriculture in the central region of Viet Nam

**Tung Thanh Le**[1]*, **Son Van Phan**[2], **Duong Thanh Nguyen**[3], **Huong Thi Hoang**[1], **Lam Trong Vu**[4]*, **Tuan Anh Le**[5], **Tuyet Thi Nguyen**[3]

1 Faculty of Social Sciences and Humanities, Duy Tan University, Da Nang, Viet Nam, 2 Faculty of Information Technology, Duy Tan University, Da Nang, Viet Nam, 3 Institute of Hospitality and Tourism, Duy Tan University, Da Nang, Viet Nam, 4 National Political Publishing House, Ha Noi, Viet Nam, 5 Faculty of Accounting, Duy Tan University, Da Nang, Viet Nam

* lethanhtung4@dtu.edu.vn (TTL); lamvutrong9@yahoo.com (LTV)

**Data Availability Statement:** All relevant data are within the paper and its Supporting information files.

## Abstract

Gender inequality and women's empowerment are two closely related issues. While the gender inequality index has been assessed by different studies, that of women's empowerment remained limited. In the present work, we attempted to evaluate the women's empowerment index by comparing it with the male partner's empowerment index in the same household. We used the Women's Empowerment in Agriculture Index (WEAI) as a framework for reference. A questionnaire was designed to interview 300 people including both men and women in the same ethnic minority household in central Vietnam. The difference in the empowerment level between men and women was assessed through five-component empowerment indicators: agricultural participation, resource ownership, financial control, social organizations participation, and time usage. The results showed that up to 70% of women were disempowered compared to only 15% of men. The binary logistic model revealed the age at first marriage, the level of children's education, education level, distance to the nearest urban area, and the number of children were associated with women's empowerment; whereas age, income, and the level of gender awareness did not show any correlation.

## Introduction

Empowerment is the ability of a group or individual to make decisions involved in resource ownership and typical activities [1, 2]. Accordingly, empowerment in agriculture is defined as the ability to own agricultural resources and make decisions on time usage and income usage; and participation in social organizations [3, 4].

Women account for about 43% of the global agricultural labor force and currently, they are contributing more than 50% of the world's food [5, 6]. Especially, in developing countries, women play a crucial role in agricultural and rural development. A study conducted recently in Ethiopia Africa showed that eighty-five percent of the population is concentrated in the agricultural sector, with women accounting for nearly half [7]. However, studies indicated that women in agriculture still face gender inequality, including gender inequality in property ownership, right of financial access, social participation, and time usage [2, 6, 8].

**Funding:** The author(s) received no specific funding for this work.

**Competing interests:** The authors have declared that no competing interests exist.

These gender gaps can reduce household income and access to basic social services such as education and healthcare [9]. Therefore, women's empowerment in agriculture is considered one of the crucial factors in ensuring global food security and promoting human development goals [2, 6, 10].

Although the role of women's empowerment in development has been addressed in several theoretical studies, empirical research on this topic is still limited. In most of the available empirical studies in rural areas, women's empowerment was often studied in a single aspect, such as economic empowerment, political empowerment, or property ownership; and very few studies have measured the composite empowerment index. One of the main reasons for this could be the difficulty in the measurement framework setup [3, 11–14]. These studies showed that women often generally have fewer rights than men to own property, including land, capital, and durable goods [15]. It was also more difficult for them to access credit and social participation than men [3, 16]. Women in Southeast Asia generally have more rights in agricultural production decisions and control of household income than those in other developing regions [17]. A number of previous studies using Logit regression models to assess empowerment-related factors revealed that educational attainment and household income were the most commonly cited variables [18–25]. Besides, age, level of gender equality, and religious awareness were noted to be significantly correlated with women's empowerment [18, 20], and some other exogenous factors such as the quality of family relationships, housing area, non-farm employment, and the distance to the nearest urban areas to a lesser extent [19].

In Vietnam, research on empowerment is often associated with research on gender equality. Therefore, there have been very few separate studies on gender empowerment in Vietnam to date. An annual national study on gender inequality mentioned that the rights of women in economics and politics showed that women often have less opportunity to make decisions in economics and politics than men. Even, in many cases, they had no right to make any decisions in their family life [26]. The main reasons for this could be cultural barriers, low educational attainment, and lack of opportunities to own productive resources such as land, capital, and property [26]. Noticeably, there are no studies in Vietnam that have measured the overall index of women's empowerment in agriculture to date.

To provide more evidence of the women's empowerment index, we conducted a study about this topic in an ethnic minority area in the central region of Vietnam. The aim of this study was to measure the overall index of women's empowerment for ethnic minorities in agriculture in this area and attempted to find out the factors associated.

## Materials and methods

### Research data

Like most other ethnic minorities all over the world, the majority of ethnic minorities in Central Vietnam live in special areas such as mountainous regions, border areas, and regions prone to frequent natural disasters, consequently, their life is always faced with many difficulties. In this study, we selected three minority communes with the highest percentage of ethnic minority populations in the Central region of Vietnam to get research samples. These are areas with a very high poverty rate, low educational attainment, and many unsound customs such as child marriage and closely consanguineous marriages.

A Dot commune (A Luoi district, Thua Thien Hue province) has 132 purely minority agricultural households. Za Hung commune (Dong Giang district, Quang Nam province) has 141 purely minority agricultural households and Tra Son commune (Tra Bong district, Quang Ngai province) has 129 purely minority agricultural households. Fifty purely agricultural households in each commune were selected for the study using a random sampling method. A

total of 150 households (300 people, including both husband and wife of each household) were included in this study.

The interviews were carried out from March to November 2018. The data collection process was divided into three stages. The first phase was performed from March to May 2018. During this period, we conducted pilot surveys and on-site training of enumerators and then modified the questionnaire. Similar to many other surveys, despite being well-trained, the interviewers still encountered difficulties in conducting and controlling the interview content. Consequently, more than half of the interviewers were eliminated after executing the first trial interview. The second phase was run from June to September 2018. For the effectiveness of data collection, we decided to include enumerators with experience in the sociological survey to collect the data. The third phase was from October till the end of November of that year. During this period, we finalized the administrative work and checked the final data.

This study is part of the socio-economic research project on the rural areas of the South-central coast of Vietnam. All the questionnaires were stamped and signed by the implementing agency. All the respondents agreed to participate in this project and answered the questionnaire.

## Methods

**Data encoding.** In the present study, we used the Women's Empowerment in Agriculture Index (WEAI) as a framework for reference. We designed a structured questionnaire that approximates the five domains in the WEAI: (i) decisions on agricultural production, (ii) owning and making decisions about resources, (iii) control and use of income, (iv) social participation, and (v) the use of time (Table 1). This questionnaire was used to interview 300 people including both men and women in the same household [3]. We construct an Empowerment Index (EI) using the same indicator weights and overall empowerment cutoff used in the WEAI." The degree of empowerment ranged from 0 to 1. Individuals reach the threshold of empowerment if he/she has a final EI larger than the cut-off value of 0.8 [3].

The answer's options for questions in each field were coded according to the Likert scale, ranging from 1 to 5 as follows: 1 = absolutely no right in decision-making; 2 = decide with someone who is not a family member; 3 = decide with a non-spouse family member; 4 = decide with the spouse, and 5 = make own decisions (highest level). The second indicator, the confidence level when making decisions was coded as follows: 1 = very unconfident, 2 = unconfident, 3 = medium, 4 = quite confident, and 5 = very confident (highest level). Other elements

**Table 1. Comparison table between WEAI's domains [3] and the study's domains.**

| WEAI's domains | | The study's domains | | Weight |
|---|---|---|---|---|
| **Domains** | **Indicators** | **Domains** | **Indicators** | |
| **Production** | Input in productive decisions | **Decisions on agricultural production** | Decision-making power | 1/10 |
| | Autonomy in production | | Confidence levels in decision-making | 1/10 |
| **Resources** | Ownership of assets | **Own and make decisions about resources** | Property ownership | 1/15 |
| | Purchase, sale, or transfer of assets | | To access loans | 1/15 |
| | Access to and decisions on credit | | Decision-making power in borrowing capital | 1/15 |
| **Income** | Control over use of income | **Control and use of income** | Control and use of income | 1/5 |
| **Leadership** | Group membership | **Social participation** | Group membership | 1/10 |
| | Speaking in public | | Confidence levels in participation | 1/10 |
| **Time use** | Workload | **Use of time** | Time to rest | 1/10 |
| | Leisure | | Time to leisure | 1/10 |

were similarly coded. The value of each empowerment factor was the average of the component indicators.

The obtained values (based on a 0–5 scale) will be converted to a 0–10 scale. For example, the average resource factor value, 3.2/5, has been converted to 6.4/10.

**The final empowerment index cut-off.** Notably, the study regions were ethnic minority communes, whose EI was significantly lower than that of other regions [3]. Therefore, besides the EI cut-off point of 0.8 that has been used by WEAI to evaluate empowerment ($\geq 0.8$), and disempowerment ($< 0.8$), in this study, we also applied an additional lower EI cut-off of 0.6 for comparison. Additionally, in the present study, the final individual-level EI was simply calculated as the weighted average of the 10 indicators in the study (Table 1). Based on the cut-off value, the respondent is defined as empowered if their empowerment score is greater than or equal to 0.8 and disempowered if their empowerment score is less than 0.8.

**Statistical analysis.** All the statistical analyses were conducted using SPSS 23.0 (SPSS Inc.). The regression analysis was performed with the dependent variable in binary (empowered = 1, disempowered = 0). Continuous variables were presented as the mean ± standard deviation. One-way analysis of variance (ANOVA) and independent sample t-test are used to test whether or not there are significant differences between the means of EI at 95% confidence level).

## Results

### Characteristics of research samples

The samples were divided almost equally by gender in which women accounted for 48.3% and men accounted for 51.7% of respondents. The average distance from respondents' houses to the nearest urban area was 19.1 km, ranging from 4 to 40 km. The average income per household/month is 3.8 million VND (about 200 USD/month). The average age at first marriage of men (20.7 years old) was higher than that of women (19.3 years old) ($p = 0.00$). Notably, the proportion of child marriage (married under 18 years old) in this area was relatively high, accounting for about 15.8% of the studied population. The rate of child marriage was more than three times higher in women (23.9%) than in men (7.6%). The average age of all respondents was 36.1 years old, in which the average age of men (38.0 years old) was higher than women (34.6 years old) (p = 0.00). The average number of years attended in school by both sexes combined was 8.7 years. The average number of years of schooling of men (9.6 years) was significantly higher than that of women (7.8 years) (p = 0.00). The average number of children was 2.9. There were no differences in the gender equality awareness index between men and women in this study (p > 0.05) (Table 2).

**Participation in agricultural production activities.** The level of empowerment in agricultural production activities was assessed via two main indicators, namely (i) decision-making power and (ii) confidence levels in decision-making. Noticeably, there was a significant difference between men and women regarding decision-making power in agricultural activities ($p = 0.00$). This rate in food crops farming and livestock raising activities was 40.0% and 33.5%; while in the non-farm economic activities and hired labor reached up to 78.7% and 89.0%, respectively. The proportion of men making their own decisions was also much higher than that of women, especially in the activities of food crop farming (46.2% vs. 11.6%), Cash crop farming (40.7% vs. 0%), livestock raising (64.8% vs. 17.4%), non–farm economic activities (41.4% vs. 5.2%), and hired-labor activities (34.5% vs. 11%) (Fig 1A).

The difference in empowerment between men and women was also clearly observed in the confidence levels in making decisions in agricultural production ($p = 0.00$). Accordingly, the percentage of women who chose the answer "very unconfident when making decisions" was

**Table 2. Characteristics of research samples.**

| Research sample | Mean ± SD | | |
|---|---|---|---|
| | Total | Men | Women |
| **Personal characteristics** | | | |
| Gender | 100% | 48.3% | 51.7% |
| Age | 36.0 ± 9.2 | 38 ± 8.8*** | 34.5 ± 9.4 |
| Education | 8.7 ± 2.8 | 9.4 ± 2.5*** | 7.8 ± 2.8 |
| The gender equality awareness index | 3.7 ± 1 | 3.6 ± 1 | 3.7 ± 1.1 |
| Age at first marriage | 20 ± 2.9 | 20.7 ± 2.7*** | 19.3 ± 3 |
| **Household characteristics** | | | |
| Distance from respondents' houses to the nearest urban area | 19.1 ± 9.7 | NA | NA |
| Income | 3.8 ± 1.6 | NA | NA |
| Number of children | 2.9 ± 1.2 | NA | NA |
| Education of children | 5.3 ± 3.7 | NA | NA |

Statistical analyses were done by Student's t-test, where ***$p < 0.0001$ indicate a significant difference between men and women groups; SD: standard deviation, NA: non-applicable.

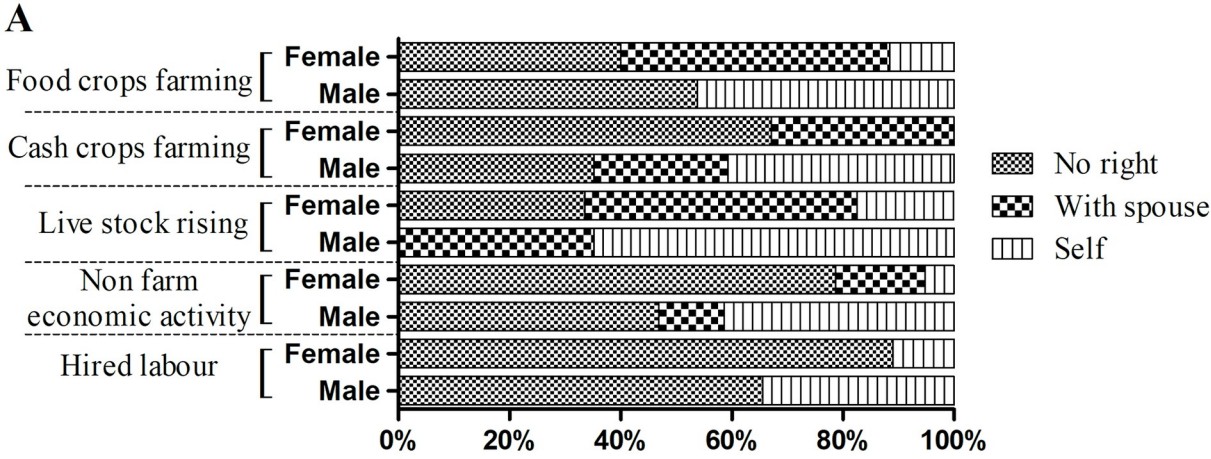

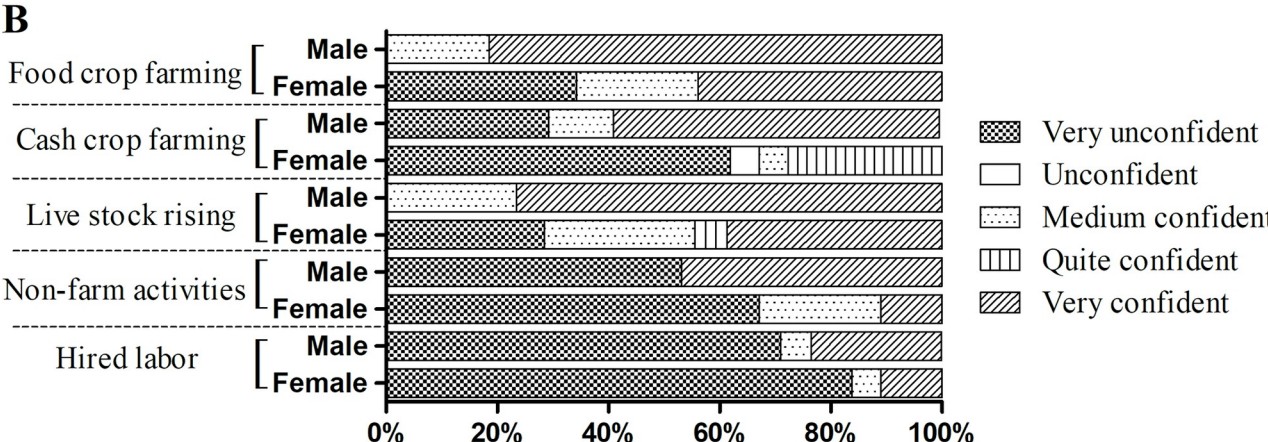

**Fig 1. The difference between men and women self-assessment in (A) Decision-making power and (B) Confidence level in making decision in agricultural production activities.**

much higher than that of men (34.2% vs. 0% for food crop farming, 61.9% vs. 29.2% for cash crop farming, 28.4% vs. 0% for livestock raising, 67.1% vs. 53.1% for non-farm activities, and 83.8% vs. 71.0% for hired labor). In parallel, the proportion of men who were "very confident" in decision-making was twice as high of women in all agricultural activities (81.4% vs. 43.9% for food crop farming, 58.6% vs. 27.7% for cash crop farming, 76.6% vs. 38.7% for livestock raising, 46.9% vs. 11% for non-agricultural activities; 23.4% vs. 11% for hired labor) (Fig 1B).

With a cut-off of 0.6, only 21.9% of the women achieved this threshold, compared to 58.4% of the men. None of the women had achieved a threshold of 0.8 compared to 11% of the men (Fig 2A).

**Own resources.** Resource ownership was assessed by three indicators, namely (i) property ownership; (ii) to access loans, and (iii) decision-making power in borrowing capital.

For content (i), our results revealed a substantial difference in property ownership between men and women (p = 0.00), especially in high-value assets. For example, up to 52.4% of men-

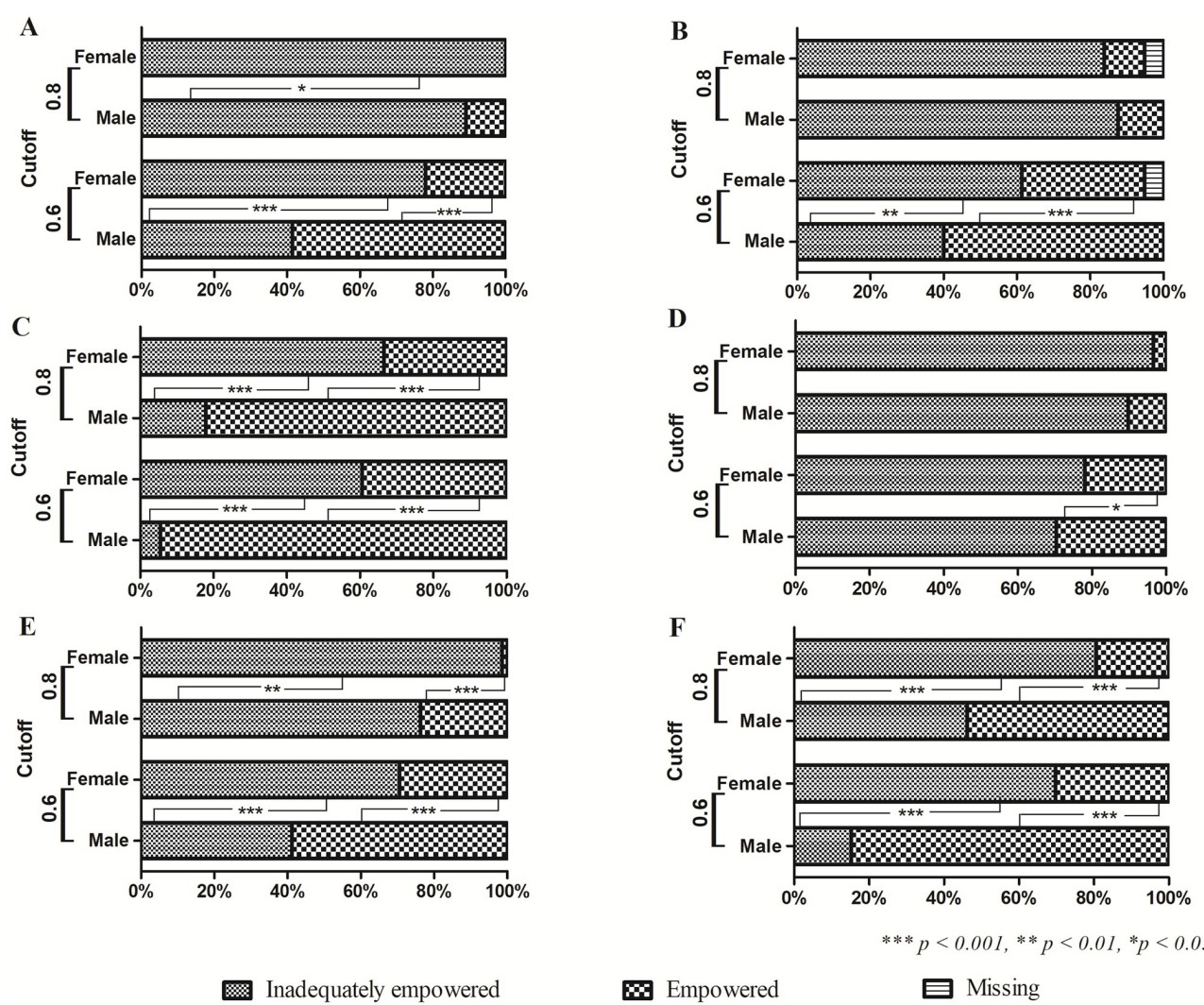

**Fig 2. The rate of reaching the threshold of empowerment of men and women at cut-off values of 0.6 and 0.8 in (A) Participation in agricultural production activities; (B) Own resources; (C) Control over the use of income; (D) Social participation; (E) Time use; (F) General empowerment in agriculture.**

**Table 3. Gender differences in owning resources (in percentage).**

| Answer | Agricultural land | | Large livestock | | Farm equipment | | House or other structures | | Large consumer durables | | Land not used for agricultural purposes | |
|---|---|---|---|---|---|---|---|---|---|---|---|---|
| | Men | Women | Men | Women | Men | Women | Men | Women | Men | Women | Men | Women |
| **No right** | 11.7 | 59.2 | 5.5 | 53.1 | 5.5 | 71.4 | 5.5 | 59.2 | 5.5 | 59.2 | 40.7 | 83.0 |
| **With other non-family member** | 0.0 | 6.7 | 0.0 | 0.0 | 0.0 | 0.0 | 0.0 | 0.0 | 0.0 | 0.0 | 0.0 | 0.0 |
| **With other family member** | 0.0 | 0.0 | 6.2 | 0.0 | 0.0 | 0.0 | 0.0 | 0.0 | 0.0 | 0.0 | 0.0 | 0.0 |
| **With spouse** | 35.9 | 34.1 | 35.9 | 40.8 | 30.4 | 28.6 | 30.4 | 34.7 | 30.3 | 34.7 | 24.1 | 17.0 |
| **Self** | 52.4 | 0.0 | 52.4 | 6.1 | 64.1 | 0.0 | 64.1 | 6.1 | 64.2 | 6.1 | 35.2 | 0.0 |
| **Total** | 100 | 100 | 100 | 100 | 100 | 100 | 100 | 100 | 100 | 100 | 100 | 100 |

owned agricultural land by themselves, while, none of the women owned this type of asset. For other assets such as castles, farm equipment, non-agricultural land, houses, and durable goods, the proportion of men who chose the answer of owning it alone (52.4, 64.1, 35.2, 64.1, and 64.2%, respectively) was substantially higher than that of women (6.1, 0.0, 0.0, 6.1, and 6.1%, respectively) (Table 3).

There was a statistically significant difference between men and women in their ability to access loans (p = 0.03). The percentage of women respondents answering that they definitely could borrow money was significantly lower than men (formal credit: 34.7% vs. 59.3%; informal credit: 40.8% vs. 55.2%). However, women were more likely than men to access a loan from non-government organizations (NGOs) (17.7% vs. 11%) but this difference was not significant (p = 0.07) (Table 4).

Regarding decision-making power in borrowing activities (iii), similar to the above two contents, there is also a big difference between men and women. In formal credit institutions, 24.1% of men made their own decisions when borrowing while none of the women did so. This ratio in the form of informal credit is 24.8% for men, compared to 6.1% for women (Table 4). There was no gender difference in decision-making for borrowing from NGOs (p > 0.05) (Table 4).

The overall value of the ownership of the resources had also shown the gender difference. Only 40% of the women achieved a cut-off of 0.6 compared to 61.3% of the men. With a cut-off of 0.8, approximately 85% of both sexes do not achieve this threshold (p > 0.05) (Fig 2B).

**Control over the use of income.** There was a statistically significant difference in the family financial control factor between men (8.5/10) and women (5.4/10) (*p* = 0.00). Specifically, the ratio of women who have no decision-making power in the major issues related to family finance usage was up to 61.3% while this ratio in men was only 5.5%. Women were often very unconfident in making big decisions (64.6%) as compared to men (5.5%). Also, the percentage of men (64.1%) who made decisions on their own was much higher than that of women (5.8%). However, the gender difference was much smaller in the proportion of men (29.7%) and women (23.8%) in making small decisions. Remarkably, only in small financial decisions, the proportion of women who responded as "very confident" (40.8%) was higher than men (35.9%) (Table 5).

With a cut-off of 0.6, only 5.5% of the women were empowered compared to 60.6% of the men. Similarly, with a cut-off of 0.8, 5.5% of the women were empowered compared to 66.5% of the men *(p = 0.00)* (Fig 2C).

**Social participation.** There are seven types of social participation listed in the questionnaire, namely agricultural organizations, forestry organizations, credit organizations, financial organizations, community support organizations, commercial organizations, charity

**Table 4. Gender differences in accessing to loans and lending decision-making power.**

| Ability to take a loan | | | Accessing and decisions on loan | | |
|---|---|---|---|---|---|
| Answer | Men | Women | Answer | Men | Women |
| NGOs (number (%)) | | | | | |
| Absolutely impossible | 111 (76.6%) | 103 (70.1%) | No right | 128 (88.3%) | 121 (82.3%) |
| Impossible | 0 (0.0%) | 0 (0.0%) | With other non-family member | 0 (0.0%) | 0 (0.0%) |
| Maybe | 9 (6.2%) | 0 (0.0%) | With other family member | 0 (0.0%) | 0 (0.0%) |
| Possible | 9 (6.2%) | 18 (12.2%) | With spouse | 8 (5.5%) | 17 (11.6%) |
| Reliably | 16 (11.0%) | 26 (17.7%) | Self | 9 (6.2%) | 9 (6.1%) |
| Total | **145 (100%)** | **147 (100%)** | Total | **145 (100%)** | **147 (100%)** |
| Formal lender (number (%)) | | | | | |
| Absolutely impossible | 59 (40.7%) | 88 (59.9%) | No right | 83 (57.2%)*** | 104 (70.7%) |
| Impossible | 0 (0.0%) | 0 (0.0%) | With other non-family member | 0 (0.0%) | 0 (0.0%) |
| Maybe | 0 (0.0%) | 0 (0.0%) | With other family member | 0 (0.0%) | 0 (0.0%) |
| Possible | 0 (0.0%) | 8 (5.4%) | With spouse | 27 (18.6%)* | 43 (29.3%) |
| Reliably | 86 (59.3%)*** | 51 (34.7%) | Self | 35 (24.1%)*** | 0 (0.0%) |
| Total | **145 (100%)** | **147 (100%)** | Total | **145 (100%)** | **147 (100%)** |
| Informal lender and friends or relatives (number (%)) | | | | | |
| Absolutely impossible | 48 (33.1%) | 61 (41.5%) | No right | 73 (50.3%) | 95 (64.6%) |
| Impossible | 0 (0.0%) | 0 (0.0%) | With other non-family member | 0 (0.0%) | 0 (0.0%) |
| Maybe | 8 (5.5%) | 0 (0.0%) | With other family member | 0 (0%) | 0 (0.0%) |
| Possible | 9 (6.2%) | 18 (12.2%) | With spouse | 36 (24.8%) | 43 (29.3%) |
| Reliably | 80 (55.2%)*** | 60 (40.8) | Self | 36 (24.8%)*** | 9 (6.1%) |
| Total | **145 (100%)** | **147 (100%)** | Total | **145 (100%)** | **147 (100%)** |

Data were presented as numbers (%). Statistical analyses were done by Student's t-test, where $^*p < 0.05$, $^{**}p < 0.001$ and $^{***}p < 0.0001$ indicate a significant difference between men and women groups; missing women: 8.

**Table 5. Gender differences on control over the use of income.**

| Decision-making on use of income | | | | | |
|---|---|---|---|---|---|
| Answer | Men | Women | Answer | Men | Women |
| Big decisions (number (%)) | | | | | |
| No right | 8 (5.5%)*** | 95 (61.3%) | Very unconfident | 8 (5.5%)*** | 95 (64.6%) |
| With other non-family member | 0 (0.0%) | 0 (0.0%) | Unconfident | 0 (0.0%) | 0 (0.0%) |
| With other family member | 0 (0.0%) | 0 (0.0%) | Medium | 0 (0.0%)** | 25 (17.0%) |
| With spouse | 44 (30.3%) | 51 (32.9%) | Quite confident | 0 (0.0%) | 0 (0.0%) |
| Self | 93 (64.1%)*** | 9 (5.8%) | Very confident | 137 (94.5%)*** | 27 (18.4%) |
| Total | **145 (100%)** | **155 (100%)** | Total | **145 (100%)** | **147 (100%)** |
| Small decisions (number (%)) | | | | | |
| No right | 9 (6.2%)** | 35 (23.8%) | Very unconfident | 9 (6.2%)*** | 44 (29.9%) |
| With other non-family member | 0 (0.0%) | 0 (0.0%) | Unconfident | 0 (0.0%) | 0 (0.0%) |
| With other family member | 0 (0.0%) | 0 (0.0%) | Medium | 68 (46.9%)** | 43 (29.3%) |
| With spouse | 93 (64.1%)* | 77 (52.4%) | Quite confident | 16 (11%)* | 0 (0.0%) |
| Self | 43 (29.7%) | 35 (23.8%) | Very confident | 52 (35.9%) | 60 (40.8%) |
| Total | **145 (100%)** | **147 (100%)** | Total | **145 (100%)** | **147 (100%)** |

Statistical analyses were done by Student's t-test; $^*p < 0.05$, $^{**}p < 0.001$ and $^{***}p < 0.0001$ indicate a significant difference between men and women groups; Missing women: 8.

organizations, and religious organizations. The level of participation of men was higher than women in all the organizations, except for religious organizations. However, these differences were marginal and not statistically significant ($p > 0.05$). There was no statistically significant difference between men and women in social participation. 21.9% of the women were empowered compared to 27.9% of the men at the cut-off of 0.6 (p > 0.05). Similarly, at the cut-off of 0.8, there was no statistically significant difference. (Fig 2D).

**Time use.** The allocation of working and leisure time was different between men and women ($p = 0.00$). Accordingly, 68.4% of women answered that their working time was from nine to thirteen hours per day, while only 4.8% of men chose this option. Most of the men in this study (94.5%) had working time from four to eight hours, meaning that women have less rest and leisure time than men. Particularly, 83.2% of women had less than two hours of rest per day while this rate in men was only 5.5%. With a cut-off of 0.6, 29.4% of the women were empowered compared to 41.2% of the men. At the cut-off of 0.8, only 1.4% of the women were empowered compared to 23.7% of the men (p = 0.00) (Fig 2E).

**Assessing the index of empowerment in agriculture.** The general assessment of the above 5 indicators revealed a big difference in the empowerment between men and women (p = 0.00). With a cut-off of 0.6, only 30.3% of the women were empowered compared to 84.8% of the men. Similarly, Only 19.4% of the women were empowered compared to 53.8% of the men at the cut-off of 0.8 (p = 0.00) (Fig 2F).

**Regression analysis of factors affecting women' empowerment in the agricultural sector.** To evaluate the empowerment's related factors, different regression models have been tested. However, the binary logistic regression model has appeared to be the most suitable model for the following reasons: (i) It is logical to use a cut-off to divide respondents into 2 groups (empowered and disempowered) which have been done for all five final empowerment domains and the final EI; (ii) As compared to other regression models, the binary regression can predict variables that can promote the probability of being empowered. This is a crucial factor in recommending specific policies for the ethnic minority areas of Vietnam.

A logistic regression analysis was performed to assess the factors associated with empowerment at both 0.6 and 0.8 cut-off values. Because the percentage of both men and women not reaching the empowerment threshold of 0.8 is very high, the regression results at this value would not be very meaningful. Therefore, we only focused on the regression results at the cut-off of 0.6. The results of the general conformity test showed *p* = 0.000, indicating that the model can explain well the hypotheses with the given independent variables. The -2 log-likelihood value was used to evaluate the suitability of the model in explaining the hypotheses. The correct prediction rate of the model was 72.3%. Among eight independent factors that were contemporaneously included in the model to evaluate their impact on the dependent variable (probability of reaching the empowered threshold), five factors were statistically significant ($p < 0.05$), namely (i) the educational level (measured by the number of years of schooling), (ii) the age at first marriage, (iii) the distance to the nearest urban area, (iv) the children's highest years of schooling, and (v) the number of children, in which (i), (ii), and (iv) showed a positive correlation while (iii) and (v) showed a negative correlation to the probability of empowerment (Fig 2F). Particularly, the higher number of years of schooling, age at first marriage, and the number of children's years of schooling, the greater chance of reaching the empowered threshold. If each of these factors increased by 1 unit (1 year) while all the other factors remained the same, the probability of reaching the empowered threshold increased by 1.22, 1.14, and 1.17 times, respectively. On the other hand, the larger the distance to the nearest urban and the number of children in the family, the lower the probability of reaching the empowered threshold. If each of these factors increased by 1 unit (1 km for the distance and 1 child for the number of children), the probability of being empowered decreased by 0.96 and 0.65 times, respectively (Table 6).

**Table 6. Regression results factors that related the probability of empowerment.**

| Factors | Variables in the equation | | | | | |
|---|---|---|---|---|---|---|
| | B | S.E. | Wald | Df | *p* value | Exp(B) |
| Age | -.033 | .031 | 1.146 | 1 | .284 | **.968** |
| Education | .199 | .074 | 7.253 | 1 | **.007** | **1.220** |
| Age at first marriage | .134 | .058 | 5.451 | 1 | **.020** | **1.144** |
| Distance to the nearest urban | -.040 | .015 | 6.683 | 1 | **.010** | **.961** |
| Children's education | .159 | .079 | 4.072 | 1 | **.044** | **1.172** |
| Income | -.150 | .337 | .197 | 1 | .657 | **.861** |
| Gender equality awareness | -.221 | .192 | 1.327 | 1 | .249 | **.802** |
| Number of children | -.428 | .170 | 6.314 | 1 | **.012** | **.652** |
| Constant | -.592 | 1.621 | .133 | 1 | .715 | **.553** |

B: Estimated coefficient; S.E: standard error of the mean; Wald: wald test meaning; Df: the degrees of freedom coefficients; *p*-value: the level of statistical significance; Exp(B): estimated coefficient weights; the number of observations: 155.

To further increase the reliability of the research results, we conducted additional analysis on the multiple regression model. This rusult did not show substantial changes in the number of variables affecting empowerment when compared to the binary regression model. However, the estimated size of the factors was different. While in the binary regression model, the education variable had the greatest association with empowerment, in the multiple regression model, the age at first marriage is the strongest influenced variable. It is also noteworthy that the household income factor has no effect on empowerment in the binary regression model, which on the other hand showed a positive effect in the multiple linear regression model (These results are not presented in the paper, but are available upon request).

## Discussion

In general, our results reflected the reality of gender differences in empowerment in agriculture-related activities in the survey areas. Particularly, the assessment of the related factors of empowerment would be an important basis for proposing policies that seek to boost the level of women's empowerment in agriculture. Similar to the previous studies in Vietnam [27], our results also showed the absolute control of men on agricultural production-related decisions. One of the reasons for this was that the rural society of Vietnam is still being influenced by Confucian views, in which men had higher positions and full power to make decisions in the family, whereas, women were obligated to perform household chores [11, 28].

In addition, our results also indicated that most of the women in the rural area have less rights than men regarding property ownership, especially for valuable properties such as farm equipment and non-agricultural land. This point had also been raised by most of the previous studies in Vietnam and some other developing countries [29, 30]. This finding could be partly explained by "culture" which was considered as one of the intangible barriers prohibiting women's property ownership. More specifically, gender stereotyping has prevented women from property ownership claims [31–34].

In concordance with most empowerment studies in developing countries, the present study also showed women have much less power than men in credit-related decisions. This observation can be explained by two main reasons: (i) in developing countries, the majority of the household heads were men, who were legally recognized and had the right to own valuable assets. Therefore, they also have the authority to mortgage these assets for loans; (ii) cultural,

legal, and economic barriers, the level of financial awareness, or differences in gender behavior also hindered women's decisions on this issue [35, 36].

Our results also show that men are still the main decision-makers in household expenditure which is different from most studies showing that women and men make joint decisions on household expenditures [11, 12]. This observation may be due to the fact that (i) the main income earners in these households are mostly men; thus the ownership and spending decisions also belong to them; (ii) the women in central Vietnam are known for their forbearance and undemanding behavior.

Among all the investigated indicators in this study, social participation is the only indicator that did not show any gender differences. However, the level of social participation of both men and women in the research area was very low. The possible reasons for such low participation in social activities are as follows: (i) the closed living space, small-scale farming practices, and self-sufficiency have created a feeling of reluctance to expand the relationships and to participate in socio-political activities; and (ii) the income of the majority of households in the study area is still very low, so they must spend more time on labor to generate income for the family.

Time use is an indicator that was clearly different between men and women. This result is also confirmed by most of the previous studies on the same topic in the rural areas of Vietnam in particular and the rural areas of other Asian countries in general [37, 38]. Women in the research areas, apart from participating in farming, also must undertake all the housework and take care of the family members. As mentioned above, due to gender role stereotyping, few men were engaged in housework with their wives. They often shun the housework and considered it as the women's duty. This is also the main reason why women do not have much rest and leisure time.

Taken together, our results showed that women have fewer rights than men in all 5 domains. The final EI also showed that nearly 70% of women in this study didn't reach the empowered threshold in agriculture. This finding is contradictory to several recent studies in other Southeast Asia regions, which indicated that women have higher rights than men in production-related activities and control the family's income [11, 13, 39]. While the education level, the distance to the nearest urban area, and the number of children are related to women's empowerment was the common findings in most of the empowerment studies [19, 21–23, 25, 35], the age at first marriage, the highest education of children detected in this study were novel factors. In particular, the age of first marriage is closely related to empowerment. Noticeably, child marriages were still common in this study area, which accounted for 15.8% of the total sample. This fact has been one of the main reasons why women in this study had less voice and power in decision-making and were dependent on their husbands and husband's family [40, 41]. Child marriages were the major factor that was directly affecting the capacity of women's empowerment in general and rural women in particular. Lastly, the highest education level of the children also had an impact on the empowerment of women. It was believed that the daily interactions in the family have changed the perception of family members, especially the right of a mother to participate and decide different family issues. These changes will encourage individuals to change gender practices, increasing the likelihood of empowerment. The factors such as age, income, and awareness of gender equality were related to the level of empowerment.

## Conclusion and policy recommendations

Firstly, in the present study, we found that in all five assessed factors, women played a lesser role than men in rural agricultural settings, across socio-demographic variables. Resource ownership, spending time, and household financial control were typical contributors to women's disempowerment.

Secondly, 70% of women in this study did not reach the empowered threshold in agriculture signifying a large gender gap in these specific geographic areas and suggesting an urgent need for specific and timely policy interventions to increase women's empowerment in these regions.

Thirdly, logistic regression results strongly suggested that, in order to reduce the gender gap in empowerment, policies need to be focused on the following issues:

(i) improving the educational level for people in remote areas, especially young women and girls. The relationship between educational attainment and children's education on the probability of empowerment indicates the need for policy interventions in this issue;

(ii) increasing the age at first marriage. Our results revealed that child marriage in mountainous regions of Vietnam is still common. Marriage at the age of under 18 plus high fertility brings multiple health implications and reduces the opportunity to access basic social services. These are the obvious barriers that hindered the empowerment of rural women today. Low education—early marriage–giving birth to too many children—poverty appears to be a "vicious circle." To be able to escape from this vicious circle it requires a "kick from outside" [42], of which livelihood support and communication for behavior change must be the key policies and done simultaneously;

(iii) promoting urbanization. The closer to the urban area, the higher probability of empowerment will be, indicating an obvious influence of the urban on the surrounding rural areas. However, urbanization is a lengthy process and its outcomes depend on multiple factors. For mountainous regions, in particular, this process would be even more difficult. The typical characteristics of the city such as diverse forms of communication, and the development of culture, were specific policy implications that can be applied in raising the awareness of rural people about the rights of themselves and the community in agriculture.

Fourthly, the results from the multiple regression model indicated that in order to improve the level of women's empowerment in rural areas, in addition to policies as suggested, it is also necessary to have policies to increase household income.

## Limitations of this study

Despite the considerable effort made, our study still has limitations. Firstly, the difficulties in sample collection made the sample size smaller than we expected. Secondly, our measurement method is completely quantitative, whereas the concept of empowerment includes qualitative factors too. Therefore, our research findings may not have fully reflected all the qualitative aspects of empowerment. We are striving to address these limitations in another study on empowerment in Vietnam.

## Supporting information

**S1 File. Questionaire.**
(PDF)

## Acknowledgments

The authors would like to thank Dr. Vinay Bharadwaj Tatipamula and Dr. Ha Thi Nguyen (Institute of Research and Development, Duy Tan University, Danang, Vietnam) for the critical reading of the manuscript.

## Author Contributions

**Conceptualization:** Tung Thanh Le, Lam Trong Vu.

**Data curation:** Son Van Phan, Duong Thanh Nguyen, Huong Thi Hoang, Tuyet Thi Nguyen.

**Formal analysis:** Tung Thanh Le, Son Van Phan, Lam Trong Vu, Tuyet Thi Nguyen.

**Investigation:** Tung Thanh Le, Duong Thanh Nguyen, Huong Thi Hoang, Lam Trong Vu, Tuyet Thi Nguyen.

**Methodology:** Tung Thanh Le, Son Van Phan, Duong Thanh Nguyen, Huong Thi Hoang.

**Resources:** Duong Thanh Nguyen.

**Supervision:** Tung Thanh Le.

**Visualization:** Tung Thanh Le, Son Van Phan.

**Writing – original draft:** Tung Thanh Le, Son Van Phan, Lam Trong Vu.

**Writing – review & editing:** Tung Thanh Le, Lam Trong Vu, Tuan Anh Le.

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
