## [Decision Letter · Decision Letter 0]

8 Jul 2022

PONE-D-21-33301Ethnic minority women’s empowerment in agriculture in the central region of Vietnam and its associated factorsPLOS ONE

Dear Dr. LE,

Thank you for submitting your manuscript to PLOS ONE. After careful consideration, we feel that it has merit but does not fully meet PLOS ONE’s publication criteria as it currently stands. Therefore, we invite you to submit a revised version of the manuscript that addresses the points raised during the review process.

Your manuscript has been assessed by two expert reviewers, whose comments are appended below. The reviewers have highlighted concerns about several aspects of the methodology and discussion, among other issues. Please ensure you respond to each point carefully in your response to reviewers document, and modify your manuscript accordingly. In addition, the manuscript requires detailed copyediting to address errors in grammar, spelling and punctuation. We strongly recommend you consult a professional scientific editing service to improve the clarity and flow of your writing.

We look forward to receiving your revised manuscript.

Kind regards,

Joseph Donlan

Editorial Office

PLOS ONE

Journal Requirements:

2. Please change "female” or "male" to "woman” or "man" as appropriate, when used as a noun (see for instance https://apastyle.apa.org/style-grammar-guidelines/bias-free-language/gender).

3. Thank you for stating the following in your Competing Interests section: "NO authors have competing interests"

Reviewers' comments:

Reviewer's Responses to Questions

**Comments to the Author**

1. Is the manuscript technically sound, and do the data support the conclusions?

Reviewer #1: Partly

Reviewer #2: No

2. Has the statistical analysis been performed appropriately and rigorously? 

Reviewer #1: Yes

Reviewer #2: No

3. Have the authors made all data underlying the findings in their manuscript fully available?

Reviewer #1: No

Reviewer #2: No

4. Is the manuscript presented in an intelligible fashion and written in standard English?

Reviewer #1: No

Reviewer #2: No

5. Review Comments to the Author

Reviewer #1: This is an interesting article. However, there are some issues with the use of the term" lose empowerment" used throughout in the introduction, results and discussion sections. I have provided some comments and suggestions for your reference.

Reviewer #2: The study describes women’s empowerment for a sample of ethnic minority women in agriculture in Central Vietnam using a composite empowerment index inspired by the Women’s Empowerment in Agriculture Index (WEAI). After constructing a composite empowerment index for each male and female respondent, a cutoff is applied to identify whether the individual is empowered or not. A binary logistic model is used to estimate the probability of being empowered, controlling for a set of individual characteristics. The study finds that age at first marriage, education level, level of children's education, distance to the nearest urban area, and the number of children are all significantly associated with increased likelihood of empowerment. The main contribution of the paper is that it describes empowerment in agriculture for a relatively understudied population – ethnic minorities in the Central Vietnam region. Overall, I think this is a worthwhile project, given the lack of research for this particular setting and sub-population. However, there several issues that should be addressed to improve the rigor of the paper.

MAJOR ISSUES

My main concern has to do with the methodology for constructing the empowerment measure and the interpretation of empowerment.

1. Confusion regarding the WEAI – The paper presents the WEAI as the tool used for measuring empowerment, but it actually does not use the WEAI. The domains and indicators may be called the same but the questions and response options are different. All the questions used Likert scales, and then the authors apply the notion of a cutoff to each individual indicator. In the WEAI, the cutoff is used to identify empowered individuals (ie, those who achieve 4 out of 5 domains or 80% of the weighted indicators are defined as empowered), whereas individual indicators each have separately defined thresholds that were informed by both theory and validation analyses. This means that the resulting indicators and composite index presented in the paper does not inherit any of the (validated) properties of the WEAI itself. Rather this is a whole new index, despite using the same domain/indicator headings and weights. At best this is “inspired” by the WEAI or is using the WEAI only as a “framework”, rather than using the WEAI methodology and tool itself. I suggest that the paper be very explicit in exactly which aspects of the WEAI methodology are used, and which parts have been modified. Otherwise readers will incorrectly assume this is using the same standardized WEAI methodology. I also suggest calling this empowerment index a different name all together to avoid confusion, and perhaps simply cite WEAI for the parts that are actually used in the paper.

2. Empirical strategy

2.1. Given that all the indicators are collected as Likert scales, converting these into binary indicators using an arbitrary cutoff is a waste of information. Instead, I suggest simply adding up the variables to come up with an aggregate scale and then use OLS to run the regressions. The use of thresholds and cutoffs in the WEAI is important for decomposition, which is no longer relevant in this case since the data (Likert scales) will not allow for WEAI-style decomposition. There are a growing number of studies that use WEAI indicators that are moving beyond simply the binary indicators (for example, Quisumbing et al 2021: https://doi.org/10.1016/j.foodpol.2020.102001). In my view this is a more appropriate indicator to use for regression analysis because it does not depend on the choice of cutoff, which the paper also points to as problematic.

2.2. Regarding the empirical specification, I wonder whether children’s education is simply picking up the age of the kids? I would also suggest using number of young children (aged <5yo) instead of number of children overall, since the care needs of young children are much greater and would arguably be more disempowering

3. Interpretation of empowerment – I was quite confused by the paper’s references to “loss of rights” throughout the paper. WEAI is primarily measuring agency, and it is indeed possible to have agency even when rights are not guaranteed. While there are indicators that can be interpreted as "rights over assets" in the WEAI, this does not extend to the whole of the empowerment index. In the WEAI’s control over income indicator, those questions are phrased as inputs over income decisions, whereas for this paper, it appears that the response options include mention of rights, but it is not clear whether that is appropriate (I would suggest sharing the questionnaire also in an Appendix). For example, the concept of rights over workload to me does not make sense, especially since the response options there appear to be a range of work hours. Rights are quite a different concept from empowerment even though they are related. I would suggest sticking to the empowerment definition that the authors refer to in lines 43-44 which is really referring to agency or the power to make important decisions.

MINOR ISSUES:

I also have additional minor comments, questions, and suggestions. Addressing these issues will significantly improve the clarity of the paper. Note that the numbers refer to line numbers in the manuscript.

1. The paper is based on observational data and therefore the regression estimates should not be interpreted as causal relationships. I suggest avoiding all mentions of “impact” and other causal language, such as in lines 311, 390, and 425.

2. Please provide clarifications on the following:

27-28: refers to “composite index” in the opening line, but not clear what is being referred to here

35-36: not sure what this means: “70% lose their empowerment in their final empowerment index”

47-50: very long sentence, hard to read; suggest putting the long lists in parentheses, or splitting this up for clarity

71: “constant variables” – constant in what sense?

91-92: is EI referring to an individual-level empowerment measure or is it like the WEAI which is reported at the program/sample level so it does not vary across individuals?

101-102: it would be useful to give more context to why this particular region was selected. Is the WEAI data collected part of a larger study?

169: Table 2: In general, household level characteristics and individual level characteristics should be presented separately. I am curious to know why distance is the only HH level characteristic of interest?

221: is this really zero? if so better to say in the text that none of the women said xyz

325: Table 6: is this for women only? Table should note the number of observations

345: maybe “male chauvinism” instead of “men chauvinism”?

340-341: “lose their rights of property ownership”: lose because of what?

390: “interaction process”: what interaction process?

399-400: “seventy percent of women in this study lost power in the final EI signified a large gender gap”: I don’t understand this point

429-432: I agree that using mixed methods is better than using only quant or qual tools by themselves. There are now qualitative protocols that are recommended for use along with the pro-WEAI for future work which I think is worth acknowledging: https://weai.ifpri.info/weai-resource-center/guides-and-instruments/.

3. There are a number of typos in the document. If the authors have the resources, I would also strongly encourage hiring a professional editor to go over the language of the paper to improve clarity and readability of the paper. Please note that PLOS ONE does not provide copyediting.

75: typo, should be “extent”

107: typo, should be “Likert”

429: typo, should be “Fifthly”

6. PLOS authors have the option to publish the peer review history of their article (what does this mean?). If published, this will include your full peer review and any attached files.

Reviewer #1: No

Reviewer #2: No

---

## [Author Response · Author response to Decision Letter 0]

8 Nov 2022

To

Editor-in-Chief,

Plos One.

Dear Editor,

With respect to your email dated Jul 8, 2022, enclosing the reviewers’ comments on the manuscript entitled “Ethnic minority women’s empowerment in agriculture in the central region of Vietnam and its associated factors” (Ref. No.: PONE-D-21-33301), we would like to thank the reviewers for the insightful review and valuable comments on our manuscript. In the revised version, we hope that we have satisfactorily addressed the comments raised by the reviewers. Here below, we provide a point-to-point answer to all remarks. All the changes have been highlighted in the manuscript in red and a clean version with line numbers was also provided. 

We hope you now will find the manuscript suitable for publication and look forward to hearing from you.

Your sincerely,

Thanh Tung Le

---

## [Decision Letter · Decision Letter 1]

17 Feb 2023

PONE-D-21-33301R1Ethnic minority women’s empowerment in agriculture in the central region of Viet Nam and its related factorsPLOS ONE

Dear Dr. LE,

Thank you for submitting your manuscript to PLOS ONE. After careful consideration, we feel that it has merit but does not fully meet PLOS ONE’s publication criteria as it currently stands. Therefore, we invite you to submit a revised version of the manuscript that addresses the points raised during the review process.

We look forward to receiving your revised manuscript.

Kind regards,

Nai-peng Tey

Guest Editor

PLOS ONE

Reviewers' comments:

Reviewer's Responses to Questions

**Comments to the Author**

1. If the authors have adequately addressed your comments raised in a previous round of review and you feel that this manuscript is now acceptable for publication, you may indicate that here to bypass the “Comments to the Author” section, enter your conflict of interest statement in the “Confidential to Editor” section, and submit your "Accept" recommendation.

Reviewer #1: All comments have been addressed

Reviewer #2: (No Response)

2. Is the manuscript technically sound, and do the data support the conclusions?

Reviewer #1: Partly

Reviewer #2: Partly

3. Has the statistical analysis been performed appropriately and rigorously? 

Reviewer #1: Yes

Reviewer #2: No

4. Have the authors made all data underlying the findings in their manuscript fully available?

Reviewer #1: No

Reviewer #2: Yes

5. Is the manuscript presented in an intelligible fashion and written in standard English?

Reviewer #1: Yes

Reviewer #2: Yes

6. Review Comments to the Author

Reviewer #1: The comments are given in the attached file. The article requires more editing for language and presentation.

Reviewer #2: The revised paper has improved considerably especially with respect to clarity. However, I have some remaining concerns.

1. First, the description of the methodology still needs further clarification:

1.1. Lines 32-33 and 84-85: “United States Feed for Future Assessment (WEAI)” – this is incorrect. I suggest revising as, “Women’s Empowerment in Agriculture Index (WEAI)”.

1.2. The methodology is described as “inheriting the five domains of empowerment” (line 84) and “inheriting WEAI’s calculations…” (line 89). I think this is still misleading because the empowerment measure that the authors use do not actually inherit the properties of the WEAI. I suggest the authors simply rephrase as: “We designed a structured questionnaire that approximates the five domains in the WEAI: (i) decisions on agricultural production, (ii) own and make decisions about resources, (iii) control and use of income, (iv) social participation, and (v) the use of time. We construct an Empowerment Index (EI) using the same indicator weights and overall empowerment cutoff used in the WEAI.”

1.3. It would help to list the actual indicators used in the study, either by adding this to Table 1 or adding a new table to the actual indicators in the Methods section. The advantage of putting all of the information in one table is that it allows for a side-by-side comparison and makes clear which indicators are similar to WEAI and which are not. For example, under the “Production” domain, the WEAI has “input in productive decisions” and “autonomy in production” indicators. In the paper, the actual indicators analyzed are “decision-making power” and “confidence levels in decision-making”. Also, where in the WEAI there is a “Leadership” domain, in this study, there is instead the “Social Participation” domain.

2. Second, I am unconvinced by the authors response regarding the rationale for converting all the indicators into binary indicators. Even the more recent WEAI studies use both binary and continuous indicators so restricting the analysis to purely the binary indicators seems rather outdated. The authors emphasized their goal of coming up with policy recommendations, but if those results are dependent on the choice of arbitrary cutoffs, then it is not clear how confident we should be about those recommendations. At the minimum I think the authors should consider implementing my suggestion and adding in a footnote to report that the results are unchanged (if it indeed does not change). However, if there are significant differences, I would certainly trust those results more than the findings from the analysis of the binary indicators.

3. Finally, here are additional suggestions to improve clarity:

3.1. Lines 269-271: How are the cutoffs applied for time use, when the questionnaire suggests that this is collected in hours? What does exceeding a 0.6 cutoff mean in practical terms?

3.2. Line 135: “loss of rights (<0.8)” – this should be “disempowerment”. The loss of rights is not the inverse of empowerment.

3.3. Lines 136-137: “…the final EI was simply calculated as the weighted average of the 5DE.” – this is not accurate since you are not calculating 5DE. I suggest rephrasing as follows: “…the final individual-level EI was simply calculated as the weighted average of the 10 indicators in the study (cite Table with the list of actual indicators used in the study).”

3.4. Lines 137-138: “Based on the cut-off value, the final EI was then rounded up into 0 (disempowered) or 1 (empowered).” – Do you mean that the respondent is defined as empowered if their empowerment score is greater than or equal to 0.8 and disempowered if their empowerment score is less than 0.8? Suggest rephrasing to clarify this point.

3.5. Lines 140-142: I think you need to clarify what the dependent variable is, which I think is whether the respondent is empowered (=1) or disempowered (=0). Is that correct? I suggest stating this explicitly in this paragraph to avoid confusion.

3.6. Line 359: “all 5 indicators” – do you mean 10 indicators? Or are there really just 5 indicators? This is not clear.

3.7. Lines 405-406: I suggest deleting the phrase in the parentheses: “(inheriting variables and calculating weights of WEAI)”

3.8. Line 405: “Our” should not be capitalized

7. PLOS authors have the option to publish the peer review history of their article (what does this mean?). If published, this will include your full peer review and any attached files.

Reviewer #1: **Yes: **Nai Peng Tey

Reviewer #2: No

---

## [Author Response · Author response to Decision Letter 1]

2 Apr 2023

30 March, 2023

To

Reviewer #1

Dear Reviewer,

We would like to thank the reviewer again for the insightful review and valuable comments on our manuscript. In the revised version, we hope that we have satisfactorily addressed the comments raised by the reviewer. 

Here below, we provide a point-to-point answer to all remarks. All the changes have been highlighted in the manuscript in red and a clean version with line numbers was also provided.

We would like to thank the reviewer again for taking the time to review our manuscript. We hope the manuscript after careful revisions meet your high standards. We are happy to respond to any further questions and comments you may have.

Your sincerely,

Thanh Tung Le

Reviewer #1

1. Consider delete and its related factors from the title.

Response: Thank you for the suggestion. The title of manuscript has now been changed as per your suggestion. Changes have been highlighted in red as the following:

Revises text: 

Ethnic minority women’s empowerment in agriculture in the central region of Viet Nam (Lines: 4 – 5).

2. My suggested changes are marked in yellow.

Response: Thank you for your suggestion. All of your mark was changed as the following: Lines: 29, 30, 31, 33, 36, 38, 41, 56, 59, 61, 62, 67, 68, 72, 74, 75, 76, 77, 122, 152, 155, 160, 206, 211, 214, 219, 279, 280, 284, 285, 320, 326, 328, 331, 340, 343, 344, 353, 355, 373, 378, 382, 384, 389, 392.

3. Need to add women's because the title of the paper is on women's empowerment

Response:

Thank you for the suggestion. The phrase “women’s empowerment” has now been changed as per your suggestion throughout the revised manuscript. Changes have been highlighted in red as the following: Lines: 29, 30,31,32,33, 41, 59, 61, 72)

4. One does not own issues related

Response: 

Thank you for pointing this out. This paragraph has now been written as per your suggestion. Changes have been highlighted in red as the following.

Revises text: 

Empowerment is the ability of a group or individual to make decisions involved in resource ownerships and typical activities [1,2]. Accordingly, empowerment in agriculture is defined as the ability to own agricultural resources and make decisions on time usage and income usage; and the level participate in social organizations [3, 4]. (Lines: 45 – 48).

5. Cite a few more studies on women's crucial roles in agriculture,

Response: Thank you for your suggestion. We added 2 cites [5, 6] on women's crucial roles in agriculture as per your suggestion. Changes have been highlighted in red as the following.

Revised text: 

Women accounts for about 43% of the global agricultural labor force who contribute to make over 50% of the world’s food [5, 6]. Especially, in developing countries, women play a crucial role in agricultural and rural developement. A study conducted recently in Ethiopia in Africa showed that eighty-five percent of the population is concentrated in the agricultural sector, with women accounted for nearly a half [7] (Lines: 49– 53).

6. State the objective of the study here.

Response: Thank you for the suggestion. The objective of the study was presented as per your suggestion. Changes have been highlighted in red as the following.

Revised text: 

To provide more evidence of the women's empowerment index, we conducted a study in ethnic minority area in the central of Vietnam. The aim of this study was to measure the overall index of women's empowerment for ethnic minority in agriculture in the central of Vietnam. Besides, we also attempted to find out factors associated with this index. (Lines: 83– 86).

7. This paragraph and Table 1 should be moved to the method section.

Response: Thank you for the suggestion. This paragraph and Table 1 was moved to the method section as per your suggestion. Changes have been highlighted from page 112 to page 130 in the manuscrift. 

8. Rewrite as “One-way analysis of variance (ANOVA) and independent sample t-test are used to test whether or not there are significant differences between the means of EI, at 95% confidence level)”

 Response: Thank you for your comment. This paragraph has now been written as per your suggestion. Changes have been highlighted in red as the following.

Revised text: 

One-way analysis of variance (ANOVA) and independent sample t-test are used to test whether or not there are significant differences between the means of EI, at 95% confidence level). (Lines: 145– 147).

9. Linea 173 to 

Response: This sentence has now been deleted as per your suggestion.

10. Show 100% to indicate reading vertically 

Response: Thank you for rasing this important question. We showed 100% as per your suggestion. Changes have been highlighted in red in table 3 (Line 226). 

11. Need to state what is this threshold 

Response: Thank you for your suggestion. This paragraph has now been written as per your suggestion. Changes have been highlighted in red as the following

Revised text: 

With a cut-off of 0.6, only 5.5 % of the women was empowered compare to 60.6% of the men. Similary, with a cut-off of 0.8, 5.5% of the women was empowered compare to 66.5 % of the men (p = 0.00) (Fig 2C) (Lines: 243– 245).

12. Instead of generation, should it be use of income?

Response: Thank you for pointing this out. Instead of generation, it must be use of income? Changes have been highlighted in red as the following

Revised text: 

Decision-making on use of income (Table 5, Line 246)

13. I don't understand this. What is from night? Please check this, and the other figures in this para.

Response: Thank you for pointing this out. Changes have been highlighted in red as the following:

Revised text: 

The allocation of working and leisure time was different between men and women (p = 0.00). Accordingly, 68.4% of women answered that their working time was from nine to thirteen hours per day, while only 4.8% of men chose this option (Lines: 262– 264).

14. Threshold of what?

Response: Thank you for your comment. Changes have been highlighted in red as the following:

Revised text: 

With a cut-off of 0.6, only 5.5 % of the women was empowered compare to 60.6% of the men. Similary, with a cut-off of 0.8, 5.5% of the women was empowered compare to 66.5 % of the men (p = 0.00) (Fig 2C) Lines: 243– 245).

15. What is this threshold?

Response: Thank you for your comment. Changes have been highlighted in red as the following:

Revised text: 

There was no statistically significant difference between men and women in the social participation. 21.9% of the women was empwered compared to 27.9% of the men at cut-off of 0.6 (p > 0.05). Similarly, at the cut-off of 0.8, there was no statistically significant difference. (Fig 2D). (Lines: 257– 260).

16. Change to Logistic regression analysis of factors affecting empowerment in the agricultural sector 

Response: Thank you for your suggestion. This paragraph has now been written as per your suggestion. Changes have been highlighted in red as the following:

Revised text:

Regression analysis of factors affecting women' empowerment in the agricultural sector (Lines: 275– 276).

17. By 4% and 35% (note one minus the odds ratio if it is less than 1. Please check.

Response: Thank you for the suggestion. The result of regression analysis was checked many time again as per your suggestion. The influence of the factors in percentage was presented in detail in the table below.

Factors Variables in the equation

 B S.E. Wald Df p value Exp(B) 

%

 Age -.033 .031 1.146 1 .284 .968 -3,2

 Education .199 .074 7.253 1 .007 1.220 22

 Age at first marriage .134 .058 5.451 1 .020 1.144 14,4

 Distance to the nearest urban -.040 .015 6.683 1 .010 .961 -3,9

 Children’s education .159 .079 4.072 1 .044 1.172 17,2

 Income -.150 .337 .197 1 .657 .861 -13,9

 Gender equality awareness -.221 .192 1.327 1 .249 .802 -19,8

 Number of children -.428 .170 6.314 1 .012 .652 -34,8

 Constant -.592 1.621 .133 1 .715 .553 -44,7

18. Why not just land?

Response: Thank you for your comment. In Viet Nam, land, especially non-agricultural land, are the most valuable asset class. Owners of this type of asset have the right to mortgage the bank to borrow capital. So we want to use it as a proxy for high value assets.

19. You need to cite a few more (besides 10) because you mentioned several recent studies.

Response: Thank you for pointing this out. Two cite was added as per your suggestion. Changes have been highlighted in red as the following:

Revised text: 

This finding is contradictory to several recent studies in other Southeast Asia regions, which indicated that women have higher rights than men in production-related activities and control the family’s income [11, 13, 39]. (Lines: 357– 359).

20. This study has several limitations which must be stasted.

Response: Thank you for your comment. The limitations of this study have been added as per your suggestion. Changes have been highlighted in red as the following:

Revised text: 

Although we have tried very hard, our research still has some limitations.

Firstly, the several difficulties in during sample collection made our sample is not large enough as desired.

Secondly, Our measurement method is completely quantitative, while empowerment is a more qualitative concept. In the process of collecting data for this study, we realized that for many issues, the collection of qualitative data through individual in-depth interviews or focus group discussions could exploit more specific information and thus provide higher quality data. 

We are trying to overcome the above limitations in an subsequent study on empowerment in ethnic minority areas, central Vietnam. (Lines: 402– 410).

Thank you very much for all your valuable comments and suggestions!

30 March, 2023

To

Reviewer #2

Dear Reviewer,

We would like to thank the reviewer again for the insightful review and valuable comments on our manuscript. In the revised version, we hope that we have satisfactorily addressed the comments raised by the reviewer. 

Here below, we provide a point-to-point answer to all remarks. All the changes have been highlighted in the manuscript in red and a clean version with line numbers was also provided.

We would like to thank the reviewer again for taking the time to review our manuscript. We hope the manuscript after careful revisions meet your high standards. We are happy to respond to any further questions and comments you may have.

Your sincerely,

Thanh Tung Le

Reviewer #2

1. Lines 32-33 and 84-85: “United States Feed for Future Assessment (WEAI)” – this is incorrect. I suggest revising as, “Women’s Empowerment in Agriculture Index (WEAI)”.

Response: Thank you for your valuable comment. This sentences has now been written as per your suggestion. Changes have been highlighted in red as the following:

Revised text: 

We used the Women’s Empowerment in Agriculture Index (WEAI) as a framework for reference (Lines: 32– 33).

In the present study, we used We used the Women’s Empowerment in Agriculture Index (WEAI) as a framework for reference (Lines: 112– 113).

2. The methodology is described as “inheriting the five domains of empowerment” (line 84) and “inheriting WEAI’s calculations…” (line 89). I think this is still misleading because the empowerment measure that the authors use do not actually inherit the properties of the WEAI. I suggest the authors simply rephrase as: “We designed a structured questionnaire that approximates the five domains in the WEAI: (i) decisions on agricultural production, (ii) own and make decisions about resources, (iii) control and use of income, (iv) social participation, and (v) the use of time. We construct an Empowerment Index (EI) using the same indicator weights and overall empowerment cutoff used in the WEAI.”

Response: Thank you for your suggestion. This paragraph has now been written as per your suggestion. Changes have been highlighted in red as the following:

Revised text: 

In the present study, we used We used the Women’s Empowerment in Agriculture Index (WEAI) as a framework for reference. We designed a structured questionnaire that approximates the five domains in the WEAI: (i) decisions on agricultural production, (ii) own and make decisions about resources, (iii) control and use of income, (iv) social participation, and (v) the use of time (Table 1). This questionnaire was used to interview 300 people including both men and women in the same household [3]. We construct an Empowerment Index (EI) using the same indicator weights and overall empowerment cutoff used in the WEAI.” The degree of empowerment ranged from 0 to 1. Individuals reach the threshold of empowerment if he/she has the final EI larger than the cut-off value of 0.8 [3]. (Lines: 113– 120).

3. It would help to list the actual indicators used in the study, either by adding this to Table 1 or adding a new table to the actual indicators in the Methods section. The advantage of putting all of the information in one table is that it allows for a side-by-side comparison and makes clear which indicators are similar to WEAI and which are not. For example, under the “Production” domain, the WEAI has “input in productive decisions” and “autonomy in production” indicators. In the paper, the actual indicators analyzed are “decision-making power” and “confidence levels in decision-making”. Also, where in the WEAI there is a “Leadership” domain, in this study, there is instead the “Social Participation” domain.

Response: Thank you for your comment. Table 1 has now been added as per your suggestion. Changes have been highlighted in red as the following:

Revised text: (Lines 130)

Table 1. Comparison table between WEAI’s domains [3] and the study’s domains. 

Weai’s domains The study’s domains Weight

Domains Indicators Domains Indicators 

Production Input in productive decisions 

Decisions on agricultural production Decision-making power 1/10

 Autonomy in production Confidence levels in decision-making 1/10

Resources Ownership of assets 

Own and make decisions about resources Property ownership 1/15

 Purchase, sale, or transfer of assets To access loans 1/15

 Access to and decisions on credit Decision-making power in borrowing capital 1/15

Income Control over use of income Control and use of income Control and use of income 1/5

Leadership Group membership 

Social participation Group membership 1/10

 Speaking in public Confidence levels in participation 1/10

Time use Workload 

Use of time Time to rest 1/10

 Leisure Time to leisure 1/10

4. Second, I am unconvinced by the authors response regarding the rationale for converting all the indicators into binary indicators. Even the more recent WEAI studies use both binary and continuous indicators so restricting the analysis to purely the binary indicators seems rather outdated. The authors emphasized their goal of coming up with policy recommendations, but if those results are dependent on the choice of arbitrary cutoffs, then it is not clear how confident we should be about those recommendations. At the minimum I think the authors should consider implementing my suggestion and adding in a footnote to report that the results are unchanged (if it indeed does not change). However, if there are significant differences, I would certainly trust those results more than the findings from the analysis of the binary indicators.

Response: Thank you for your suggestion. The analysis results by multiple regression model has now been added in the foot note as per your suggestion. Changes have been highlighted in red as the following:

Revised text: (page 16)

The multiple regression model analysis did not show substantial changes in the number of variables affecting empowerment when compared to the binary regression model. However, the weight order of the influenced factors was different between two models. While in the binary regression model, the education variable had the greatest impact on empowerment, in the multiple regression model, the age at first marriage is the strongest influenced variable. It is also noteworthy that the household income factor has no effect on empowerment in the binary regression model, which on the other hand showed a positive effect in the multiple linear regression model. These results indicated that in order to improve the level of women's empowerment in rural areas, besides the the policies as suggested in this paper, it is also necessary to have policies to increase household income.

5. Lines 269-271: How are the cutoffs applied for time use, when the questionnaire suggests that this is collected in hours? What does exceeding a 0.6 cutoff mean in practical terms?

Response: Thank you for your comment. In this study, the total 24-hour time spent on all activities was categorized into three main areas: working time, rest time (which included sleep, socializing, and meeting other people), and leisure time (used for activities such as watching television, reading, and listening to the radio). The empowerment in use the time was only assessed based on two indicators: rest time and leisure time. To ensure consistency in the analysis, we converted the duration of these activities into five levels on the likert scale. For example, 0 to 3 hours was assigned level 1, over 3 to 5 hours was assigned level 2, over 5 to 7 hours was assigned level 3, over 7 to 10 hours was assigned level 4, and 10 hours or more was assigned level 5. The average value was then calculated from the converted values. An average value exceeding 0.6 indicates that an individual was empowered on time use.

6. Line 135: “loss of rights (<0.8)” – this should be “disempowerment”. The loss of rights is not the inverse of empowerment.

Response: Thank you for the suggestion. This sentence has now been written as per your suggestion. Changes have been highlighted in red as the following:

Revises text: 

Therefore, besides the EI cut-off point of 0.8 that has been used by WEAI to evaluate empowerment (≥ 0.8), and disempowerment (< 0.8), in this study, we also applied an additional lower EI cut-off of 0.6 for comparison. (Lines: 135– 137).

7: Lines 136-137: “…the final EI was simply calculated as the weighted average of the 5DE.” – this is not accurate since you are not calculating 5DE. I suggest rephrasing as follows: “…the final individual-level EI was simply calculated as the weighted average of the 10 indicators in the study (cite Table with the list of actual indicators used in the study).”

Response: Thank you for the suggestion. This sentence has now been written as per your suggestion. Changes have been highlighted in red as the following:

Revises text: 

Additionally, in the present study, the final individual - level EI was simply calculated as the weighted average of the 10 indicators in the study (Table 1) (Lines: 137– 139).

8. Lines 137-138: “Based on the cut-off value, the final EI was then rounded up into 0 (disempowered) or 1 (empowered).” – Do you mean that the respondent is defined as empowered if their empowerment score is greater than or equal to 0.8 and disempowered if their empowerment score is less than 0.8? Suggest rephrasing to clarify this point.

Response: Thank you for the comment. This sentence has now been written as per your suggestion. Changes have been highlighted in red as the following:

Revises text: 

Based on the cut-off value, the respondent is defined as empowered if their empowerment score is greater than or equal to 0.8 and disempowered if their empowerment score is less than 0.8 (Lines: 139– 141).

9. Lines 140-142: I think you need to clarify what the dependent variable is, which I think is whether the respondent is empowered (=1) or disempowered (=0). Is that correct? I suggest stating this explicitly in this paragraph to avoid confusion.

Response: Thank you for the comment. This sentence has now been written as per your suggestion. Changes have been highlighted in red as the following:

Revises text: 

All the statistical analyses were conducted using SPSS 23.0 (SPSS Inc.). The regression analysis was performed with the dependent variable in binary (empowered =1, disempowered =0). (Lines: 143– 144).

10. Line 359: “all 5 indicators” – do you mean 10 indicators? Or are there really just 5 indicators? This is not clear.

Response: This sentence has now been written as per your suggestion. Changes have been highlighted in red as the following:

Revised text: 

Taken together, our results showed that women have much lesser rights than men in all 5 domains. (Lines: 355 – 356).

11. Lines 405-406: I suggest deleting the phrase in the parentheses: “(inheriting variables and calculating weights of WEAI)”

Response: Thank you for the comment. This phrase was deleted as per your suggestion. 

12. Line 405: “Our” should not be capitalized

Response: Thank you for the comment. This word was edited as per your suggestion. Changes have been highlighted in red as the following:

Revised text: 

our measurement method is completely quantitative, while empowerment is a more qualitative concept. (Lines: 405 – 406).

Thank you very much for all your valuable comments and suggestions!

---

## [Decision Letter · Decision Letter 2]

7 May 2023

PONE-D-21-33301R2Ethnic minority women’s empowerment in agriculture in the central region of Viet NamPLOS ONE

Dear Dr. LE,

Thank you for submitting your manuscript to PLOS ONE. After careful consideration, we feel that it has merit but does not fully meet PLOS ONE’s publication criteria as it currently stands. Therefore, we invite you to submit a revised version of the manuscript that addresses the points raised during the review process.

We look forward to receiving your revised manuscript.

Kind regards,

Nai-peng Tey, Ph.D

Guest Editor

PLOS ONE

Journal Requirements:

Additional Editor Comments:

Dear authors,

One reviewer has submitted her report and requested some corrections to your manuscript. Please address her comments and make the necessary change, and re-submit your paper. If you do not want to make any changes to the following five comments by the reviewer, please justify.

1. The title does not fully reflect the content of the study. The title should reflect gender differences in relation to agricultural land tenure and decisions about agricultural activities. That's because in all sections, including results, discussions and conclusions, there are clearly presented differences between men and women related to these activities. I therefore suggest that the abstract part also be rewritten accordingly.

2. Introduction should be provided with information about Vietnam's social context on women's and men's empowerment in terms of property ownership, power between men and women in social life and employment, especially in terms of agricultural land use. In addition, policies on agricultural land use rights are provided in more detail so that readers can better understand this field in Vietnam. From there, give the significance of the research when practiced in Vietnam.

3. Research data: Authors need to supplement information on economic, social, cultural characteristics, agricultural land use rates, occupational patterns in the localities to collect data to help readers understand why research to decide on the selection of information collection in these localities.

4. Data analysis: I think the results of the logistic regression model are not strong enough. Because factors from the husband's side can play very important role related to women' empowerment in the agricultural sector, such as husband's education, age gap between husband and wife, and husband's occupation. In Vietnamese culture, women are required to be subordient for 3 men in their life, including father's subordient when unmarried, husband's subordient when married and son's subordient when husband died. Therefore, the regression model is flawed when it does not take into account the husband's characteristics in women's empowerment.

5. Policy recommendations: As I mentioned in the analysis of the results, because the research is lacking in taking into account the husband's characteristics in women's empowerment. Therefore, suggesting policies of this study may be flawed if only focusing on policies for women, while Vietnamese women are still heavily influenced by their husbands and in-laws.

Thank you and best regards.

Nai Peng TEY

Reviewers' comments:

Reviewer's Responses to Questions

**Comments to the Author**

1. If the authors have adequately addressed your comments raised in a previous round of review and you feel that this manuscript is now acceptable for publication, you may indicate that here to bypass the “Comments to the Author” section, enter your conflict of interest statement in the “Confidential to Editor” section, and submit your "Accept" recommendation.

Reviewer #1: All comments have been addressed

Reviewer #2: All comments have been addressed

Reviewer #3: (No Response)

2. Is the manuscript technically sound, and do the data support the conclusions?

Reviewer #1: Yes

Reviewer #2: Yes

Reviewer #3: Partly

3. Has the statistical analysis been performed appropriately and rigorously? 

Reviewer #1: Yes

Reviewer #2: Yes

Reviewer #3: No

4. Have the authors made all data underlying the findings in their manuscript fully available?

Reviewer #1: No

Reviewer #2: Yes

Reviewer #3: No

5. Is the manuscript presented in an intelligible fashion and written in standard English?

Reviewer #1: Yes

Reviewer #2: Yes

Reviewer #3: Yes

6. Review Comments to the Author

Reviewer #1: Thank you for addressing my earlier comments. However, you should edit the manuscript carefully. For example,

Lines 47-48 ... and the level participate in social organizations [3, 4]. - delete the level

Lines 490-0 : who contribute to make over 50% of the world’s food [- change to "contributing more than 50% of the world's food.

Line 84-85. in the central of Vietnam - should be in the central region (and where applicable)

The section on limitations requires extensive editing.

Reviewer #2: I am satisfied with the author’s revisions and response to my comments. I offer some final copyediting suggestions below:

Line 49: “Women accounts” – replace with “Women account”

Line 53: “women accounted for nearly a half” – delete “a”, “women accounted for nearly half”

Line 65: “fewer rights than men to property ownership” – replace with: “fewer rights than men to own property”

Line 69: “regression model” – replace with plural “regression models”

Line 75: “rural areas and ethnic minority” – replace with “rural areas and among ethnic minorities”

Lines 83-84: “study in ethnic minority area in the central of Vietnam” – replace with: “study in an ethnic minority area in central Vietnam”

Lines 85-86: “ethnic minority in agriculture in the central of Vietnam. Besides, we also attempted to find…” – replace with: “ethnic minorities in agriculture in this area and attempted to find…”

Line 89: “three minorities communes” – replace with: “three minority communes”

Line 105: “phase was the last two months, from October” – replace with: “phase was from October”

Line 121-122: “The answer’s options for questions in each field were coded according to the Likert scale, and coded with a number from 1 to 5, as follows:” – replace with “The answer’s options for questions in each field were coded according to the Likert scale, ranging from 1 to 5 as follows:”

Lines 124-125: “The second indicator, the confidence level when making decisions:” – replace with: “The second indicator, the confidence level when making decisions were coded as follows:”

Line 129: “has been converted to 6.4/1.” – replace with “has been converted to 6.4/10.”

Line 130, Table 1 heading: “Weai domains” – replace with “WEAI domains”

Lines 154 and 158: I suggest replacing “yo” with “years old” for clarity and consistency

Line 173: “was 40 and 33.5%” – suggest adding % as follows: “was 40% and 33.5%”

Line 174: “up to 78.7 and 89.0%” – suggest adding % as follows: “up to 78.7% and 89.0%”

Line 280: “has been done” – replace with “which has been done”

Footnote on Line 283: “weight order” is not clear, perhaps you mean “estimated size of the coefficients”?

Footnote on Line 283: “impact on empowerment” – replace with: “association with empowerment”

Footnote on Line 283: “rural areas, besides the the policies” – replace with “rural areas, in addition to policies”

Footnote on Line 283: I suggest adding this final sentence in the footnote: “These results are not presented in the paper, but are available upon request.”

Line 318: “confucius’ views” – should this be capitalized “Confucius’ views” or perhaps replaced with “Confucian views”?

Line 322: “less rights than men of property ownership” – replace with “less rights than men regarding property ownership” OR “less rights than men to own property”

Line 339: I’m not sure that “endurance” is the right word here. Suggest rephrasing or deleting “endurance and”

Line 348: “The time use is” – replace with “Time use is”

Lines 353-354: “considered it as a women’s duty” – replace with: “considered it as the women’s duty”

Line 355: “women have much lesser rights” – replace with: “women have fewer rights”

Line 365: “had lower voices” – replace with: “had less voice”

Line 377: “typical elements” – replace with: “typical contributors to women’s disempowerment”

Line 392: “it is required” – replace with “it requires”

Reviewer #3: Reviewer comments:

Ethnic minority women’s empowerment in agriculture in the central region of Viet Nam

Research plays an important role in assessing differences in the empowerment of men and women in the ownership and use of agricultural land. Research to fill in gaps in land use research and policy formulation in Vietnam. The study still has a few points that need to be added and improved as follows:

1. The title does not fully reflect the content of the study. The title should reflect gender differences in relation to agricultural land tenure and decisions about agricultural activities. That's because in all sections, including results, discussions and conclusions, there are clearly presented differences between men and women related to these activities. I therefore suggest that the abstract part also be rewritten accordingly.

2. Introduction should be provided with information about Vietnam's social context on women's and men's empowerment in terms of property ownership, power between men and women in social life and employment, especially in terms of agricultural land use. In addition, policies on agricultural land use rights are provided in more detail so that readers can better understand this field in Vietnam. From there, give the significance of the research when practiced in Vietnam.

3. Research data: Authors need to supplement information on economic, social, cultural characteristics, agricultural land use rates, occupational patterns in the localities to collect data to help readers understand why research to decide on the selection of information collection in these localities.

4. Data analysis: I think the results of the logistic regression model are not strong enough. Because factors from the husband's side can play very important role related to women' empowerment in the agricultural sector, such as husband's education, age gap between husband and wife, and husband's occupation. In Vietnamese culture, women are required to be subordient for 3 men in their life, including father's subordient when unmarried, husband's subordient when married and son's subordient when husband died. Therefore, the regression model is flawed when it does not take into account the husband's characteristics in women's empowerment.

5. Policy recommendations: As I mentioned in the analysis of the results, because the research is lacking in taking into account the husband's characteristics in women's empowerment. Therefore, suggesting policies of this study may be flawed if only focusing on policies for women, while Vietnamese women are still heavily influenced by their husbands and in-laws.

7. PLOS authors have the option to publish the peer review history of their article (what does this mean?). If published, this will include your full peer review and any attached files.

Reviewer #1: **Yes: **Nai Peng TEY

Reviewer #2: No

Reviewer #3: No

---

## [Author Response · Author response to Decision Letter 2]

28 May 2023

28 May 2023

To 

Guest Editor,

Plos One.

Dear Guest Editor,

With respect to your email dated May 7, 2023, enclosing the reviewers’ comments on the manuscript entitled “Ethnic minority women’s Empowerment in Agriculture in the central region of Vietnam” (Ref. No.: PONE-D-21-33301R2). 

We would like to thank the Editor and reviewers once again for the insightful review and valuable comments on our manuscript. In this revised version, in addition to addressing all the requests from the editor and reviewers, the manuscript has also been further proofread by experienced researchers. We have also been meticulous in correcting any possible spelling errors in the manuscript. We hope that we have satisfactorily addressed the comments raised by the Editor and reviewers. 

We also hope you now will find the manuscript suitable for publication and look forward to hearing from you.

Yours sincerely,

Thanh Tung Le

1. Editor’s comments

1. The requirement for submitting the revised manuscript:

Response: Thank you for your requirement. Our revised manuscript included the following items as required: 

 1. A rebuttal letter that responds to each point raised by the academic editor and reviewers. This file is labeled "Response to Reviewers".

2. A marked-up copy of the manuscript that highlights changes made to the original version. This file is labeled 'Revised Manuscript with Track Changes'.

3. An unmarked version of the revised paper without tracked changes. This file is labeled "Manuscript".

Response: Thank you for your requirement. We have thoroughly checked all 42 references in the manuscript. We checked each reference by accessing the original websites of the journals to verify their publication status up to the present time. After careful examination, we found the following:

Firstly, all the references are complete and accurate.

Secondly, all the citations correspond accurately to the content and page numbers.

Thirdly, none of our references have been retracted as of the current time.

2. Response to Additional Editor Comments

1. The title does not fully reflect the content of the study. The title should reflect gender differences in relation to agricultural land tenure and decisions about agricultural activities. That's because in all sections, including results, discussions and conclusions, there are clearly presented differences between men and women related to these activities. I therefore suggest that the abstract part also be rewritten accordingly.

Response: Thank you very much for the suggestion. Our research focuses on evaluating gender differences. More specifically, the gender disparity in the index of empowerment, where agricultural land use rights and decision-making in agricultural activities are two out of the ten factors that constitute the overall empowerment level.

Because of this reason, our originally planned title was "Gender Gap in Empowerment in Agriculture among ethnic minorities in Central Vietnam." However, after completing the manuscript, we realized that the majority of women were not empowered. Therefore, we wanted to adjust the title of the study to focus more on women's empowerment. That's the reason, in the first manuscript submitted to the journal, our research was titled "Level of Women' empowerment in Agriculture of Ethnic Minorities of Central Vietnam and Influencing Factors". 

After two rounds of review, both reviewers, especially the first reviewer, suggested that we consider adjusting the title of the study to "Ethnic minority women’s Empowerment in Agriculture in the central region of Viet Nam".

We realized this to be the most accurate title that reflects the aim of this research. Therefore, We would like to keep the title of the study as it is in the manuscript.

Following the reviewer's suggestion, the study's abstract was adjusted to clarify the gender gap in empowerment further. Changes have been highlighted in red as the following:

Revised text: 

Gender inequality and women's empowerment are two closely related issues. While the gender inequality index has been assessed by different studies, that of women's empowerment remained limited. In the present work, we attempted to evaluate the women's empowerment index by comparing it with the male partner's empowerment index in the same household. We used the Women’s Empowerment in Agriculture Index (WEAI) as a framework for reference. A questionnaire was designed to interview 300 people including both men and women in the same ethnic minority household in central Vietnam. The difference in the empowerment level between men and women was assessed through five-component empowerment indicators: agricultural participation, resource ownership, financial control, social organizations participation, and time usage. The results showed that up to 70% of women were disempowered compared to only 15% of men. The binary logistic model revealed the age at first marriage, the level of children's education, education level, distance to the nearest urban area, and the number of children were associated with women's empowerment; whereas age, income, and the level of gender awareness did not show any correlation. (Lines 29 – 42). 

2. Introduction should be provided with information about Vietnam's social context on women's and men's empowerment in terms of property ownership, power between men and women in social life and employment, especially in terms of agricultural land use. In addition, policies on agricultural land use rights are provided in more detail so that readers can better understand this field in Vietnam. From there, give the significance of the research when practiced in Vietnam.

Response: Thank you for the suggestion. The introduction of manuscript has now been changed as per your suggestion. Changes have been highlighted in red as the following:

Revised text: 

In Vietnam, research on empowerment is often associated with research on gender equality. Therefore, there have been very few separate studies on gender empowerment in Vietnam to date. An annual national study on gender inequality mentioned that the rights of women in economics and politics showed that women often have less opportunity to make decisions in economics and politics than men. Even, in many cases, they had no right to make any decisions in their family life [26]. The main reasons for this could be cultural barriers, low educational attainment, and lack of opportunities to own productive resources such as land, capital, and property [26]. Noticeably, there are no studies in Vietnam that have measured the overall index of women's empowerment in agriculture to date (Lines 75 – 83). 

3. Research data: Authors need to supplement information on economic, social, cultural characteristics, agricultural land use rates, occupational patterns in the localities to collect data to help readers understand why research to decide on the selection of information collection in these localities.

Response: Thank you for the suggestion. The research data of manuscript has now been changed as per your suggestion. Changes have been highlighted in red as the following:

Revised text: 

Like most other ethnic minorities all over the world, the majority of ethnic minorities in Central Vietnam live in special areas such as mountainous regions, border areas, and regions prone to frequent natural disasters, consequently, their life is always faced with many difficulties. In this study, we selected three minority communes with the highest percentage of ethnic minority populations in the Central region of Vietnam to get research samples. These are areas with a very high poverty rate, low educational attainment, and many unsound customs such as child marriage and closely consanguineous marriages. (Lines 90 – 96). 

4. Data analysis: I think the results of the logistic regression model are not strong enough. Because factors from the husband's side can play very important role related to women' empowerment in the agricultural sector, such as husband's education, age gap between husband and wife, and husband's occupation. In Vietnamese culture, women are required to be subordient for 3 men in their life, including father's subordient when unmarried, husband's subordient when married and son's subordient when husband died. Therefore, the regression model is flawed when it does not take into account the husband's characteristics in women's empowerment.

Response: thank you very much for your valuable question. To answer the question, we would like to provide the following response:

Firstly, during the ideation and discussion process to conduct this research, the authors in our research team were well aware of the factors from the husband's side that could influence the level of empowerment of the wife. Therefore, in the initial pilot questionnaire, we included several factors such as occupation, habits, and the extent of the husband's involvement in non-agricultural activities. However, after completing the first round of the pilot survey, we noticed that in the particular minority ethnic area where we conducted the study, most of the characteristics we included in the questionnaire were highly homogeneous across almost all respondents. For example, 100% of the husbands were engaged in agricultural work, and even 100% of the wives were also involved in agricultural work and household chores. Moreover, this was a very long questionnaire that had to be administered at nearly the same time (consecutively) to both the wife and the husband within the same household. Therefore, to simplify the questionnaire, we decided to eliminate all non-statistically significant factors, including the occupation of both the wife and the husband.

Secondly, the remaining two factors related to the husband's characteristics in our final version of the questionnaire were age and education level. Prior to data analysis, we expected the husband's education level to be a significant variable. However, when analyzed using regression models (both multiple regression and binary logistic regression models), neither of these two factors had any influence on the level of women's empowerment.

Thirdly, as mentioned in previous responses, this is a study on empowerment in a specific minority ethnic area of Vietnam. Therefore, we want to focus on the main factors that contribute to suggesting appropriate policies. That is why our regression model only includes variables as presented in the manuscript.

Finally, it is possible that these findings are specific to a particular minority ethnic area, such as the one we studied. In another recent study, we conducted in a different non-minority ethnic area, the education level of the husband had a clear impact on the level of women's empowerment.

5. Policy recommendations: As I mentioned in the analysis of the results, because the research is lacking in taking into account the husband's characteristics in women's empowerment. Therefore, suggesting policies of this study may be flawed if only focusing on policies for women, while Vietnamese women are still heavily influenced by their husbands and in-laws.

Response: thank you very much for your comment. As mentioned in our response to question 4, our focus is solely on providing policy recommendations based on the outcomes of the regression model we conducted. We intend to incorporate additional policy recommendations in an other study, as we identify further factors that influence the level of women's empowerment.

Thank you very much for all your valuable comments and suggestions!

28 May 2023

To

Reviewer #1

Dear Reviewer,

We would like to thank the reviewer again for the insightful review and valuable comments on our manuscript. In the revised version, we hope that we have satisfactorily addressed the comments raised by the reviewer. 

Here below, we provide a point-to-point answer to all remarks. All the changes have been highlighted in the manuscript in red and a clean version with line numbers was also provided. We have also been meticulous in correcting any possible spelling errors in the manuscript. We hope that we have satisfactorily addressed the comments raised by the Editor and reviewers. 

We would like to thank the reviewer again for taking the time to review our manuscript. We hope the manuscript after careful revisions meet your high standards. We are happy to respond to any further questions and comments you may have.

Yours sincerely,

Thanh Tung Le

Reviewer #1

1. Lines 47-48 ... and the level participate in social organizations [3, 4]. - delete the level

Response: Thank you for the suggestion. This phrase has now been changed as per your suggestion. Changes have been highlighted in red as the following:

Revises text: 

make decisions on time usage and income usage; and participation in social organizations [3, 4] (Lines: 47 – 48).

2. Lines 49-50 : who contribute to make over 50% of the world’s food [- change to "contributing more than 50% of the world's food.

Response: Thank you for the suggestion. This paragraph has now been changed as per your suggestion. Changes have been highlighted in red as the following:

Revises text: 

Women account for about 43% of the global agricultural labor force and currently, they are contributing more than 50% of the world's food [5, 6] (Lines: 49 – 50).

3. Line 84-85 in the central of Vietnam - should be in the central region (and where applicable)

Response: Thank you for the suggestion. This paragraph has now been changed as per your suggestion. Changes have been highlighted in red as the following:

Revises text: 

To provide more evidence of the women's empowerment index, we conducted a study about this topic in an ethnic minority area in the central region of Vietnam (Lines: 84 – 85).

4. The section on limitations requires extensive editing.

Response: Thank you for the requirement. The section on the limitations of this study has now been changed as per your requirement. Changes have been highlighted in red as the following:

Revises text: 

Despite the considerable effort made, our study still has limitations. Firstly, the difficulties in sample collection made the sample size smaller than we expected. Secondly, our measurement method is completely quantitative, whereas the concept of empowerment includes qualitative factors too. Therefore, our research findings may not have fully reflected all the qualitative aspects of empowerment. We are striving to address these limitations in another study on empowerment in Vietnam (Lines: 412 – 417).

Thank you very much for all your valuable comments and suggestions!

28 May 2023

To

Reviewer #2

Dear Reviewer,

We would like to thank the reviewer again for the insightful review and valuable comments on our manuscript. In the revised version, we hope that we have satisfactorily addressed the comments raised by the reviewer. 

Here below, we provide a point-to-point answer to all remarks. All the changes have been highlighted in the manuscript in red and a clean version with line numbers was also provided. We have also been meticulous in correcting any possible spelling errors in the manuscript. We hope that we have satisfactorily addressed the comments raised by the Editor and reviewers. 

We would like to thank the reviewer again for taking the time to review our manuscript. We hope the manuscript after careful revisions meet your high standards. We are happy to respond to any further questions and comments you may have.

Your sincerely,

Thanh Tung Le

Reviewer #2

1. Line 49: “Women accounts” – replace with “Women account”

Response: Thank you for your comment. This phrase has now been replaced as per your suggestion. Changes have been highlighted in red as the following: 

Revised text: 

Women account for about 43% of the global agricultural labor force and currently, they are contributing more than 50% of the world's food [5, 6] (Lines: 49 – 50).

2. Line 53: “women accounted for nearly a half” – delete “a”, “women accounted for nearly half”

Response: Thank you for your suggestion. This word has now been deleted as per your suggestion. Changes have been highlighted in red as the following:

Revised text: 

A study conducted recently in Ethiopia Africa showed that eighty-five percent of the population is concentrated in the agricultural sector, with women accounting for nearly half (Lines: 52 – 53).

3. Line 65: “fewer rights than men to property ownership” – replace with: “fewer rights than men to own property”

Response: Thank you for your comment. This phrase has now been replaced as per your suggestion. Changes have been highlighted in red as the following:

Revised text: 

These studies showed that women often generally have fewer rights than men to own property (Lines: 64 – 65).

4. Line 69: “regression model” – replace with plural “regression models”

Response: Thank you for your suggestion. This phrase has now been replaced as per your suggestion. Changes have been highlighted in red as the following:

Revised text: 

A number of previous studies using Logit regression models to assess empowerment (Lines: 68 – 69).

5. Line 75: “rural areas and ethnic minority” – replace with “rural areas and among ethnic minorities”

Response: Thank you for your suggestion. According to Additional Editor Comments, we have edited and added information about studies in Vietnam regarding empowerment. Therefore, this sentence has been adjusted. Changes have been highlighted in red as the following:

Revised text: 

In Vietnam, research on empowerment is often associated with research on gender equality. Therefore, there have been very few separate studies on gender empowerment in Vietnam to date (Lines: 75 – 76).

6. Lines 83-84: “study in ethnic minority area in the central of Vietnam” – replace with: “study in an ethnic minority area in central Vietnam”

Response: Thank you for the suggestion. This phrase has now been replaced as per your suggestion. Changes have been highlighted in red as the following:

Revises text: 

a study about this topic in an ethnic minority area in the central region of Vietnam (Lines: 84 – 85).

7: Lines 85-86: “ethnic minority in agriculture in the central of Vietnam. Besides, we also attempted to find…” – replace with: “ethnic minorities in agriculture in this area and attempted to find…”

Response: Thank you for the suggestion. This sentence has now been written as per your suggestion. Changes have been highlighted in red as the following:

Revises text: 

The aim of this study was to measure the overall index of women's empowerment for ethnic minorities in agriculture in this area and attempted to find out the factors associated (Lines: 86 – 87).

8. Line 89: “three minorities communes” – replace with: “three minority communes”

Response: Thank you for the suggestion. This phrase has now been replaced as per your suggestion. Changes have been highlighted in red as the following:

Revises text: 

we selected three minority communes with the highest percentage of ethnic minority populations in the Central region to get research samples (Lines: 92 – 93).

9. Line 105: “phase was the last two months, from October” – replace with: “phase was from October”

Response: Thank you for the suggestion. This phrase has now been replaced as per your suggestion. Changes have been highlighted in red as the following:

Revises text: 

The third phase was from October till the end of November of that year (Line: 112).

10. Line 121-122: “The answer’s options for questions in each field were coded according to the Likert scale, and coded with a number from 1 to 5, as follows:” – replace with “The answer’s options for questions in each field were coded according to the Likert scale, ranging from 1 to 5 as follows:”

Response: Thank you for the suggestion. This sentence has now been written as per your suggestion. Changes have been highlighted in red as the following:

Revised text: 

The answer’s options for questions in each field were coded according to the Likert scale, ranging from 1 to 5 as follows: (Lines: 128 - 129).

11. Lines 124-125: “The second indicator, the confidence level when making decisions:” – replace with: “The second indicator, the confidence level when making decisions were coded as follows:”

Response: Thank you for the suggestion. This sentence has now been written as per your suggestion. Changes have been highlighted in red as the following:

Revised text: 

The second indicator, the confidence level when making decisions was coded as follows: (Lines: 131 - 132).

12. Line 129: “has been converted to 6.4/1.” – replace with “has been converted to 6.4/10.”

Response: Thank you for the comment. This number has now been edited as per your suggestion. Changes have been highlighted in red as the following:

Revised text: 

For example, the average resource factor value, 3.2/5, has been converted to 6.4/10 (Lines: 135 - 136).

13. Line 130, Table 1 heading: “Weai domains” – replace with “WEAI domains”

Response: Thank you for the comment. This phrase has now been edited as per your suggestion. Changes have been highlighted in red as the following:

Revised text: 

WEAI’s domains ( table 1 – line 137)

14. Lines 154 and 158: I suggest replacing “yo” with “years old” for clarity and consistency

Response: Thank you for the comment. There words has now been replaced as per your suggestion. Changes have been highlighted in red as the following:

Revised text: 

years old (Lines: 161, 165, 166).

15. Line 173: “was 40 and 33.5%” – suggest adding % as follows: “was 40% and 33.5%”

Response: Thank you for the comment. This number has now been added as per your suggestion. Changes have been highlighted in red as the following:

Revised text: 

This rate in food crops farming and livestock raising activities was 40.0% and 33.5%; (Lines: 180 - 181).

16. Line 174: “up to 78.7 and 89.0%” – suggest adding % as follows: “up to 78.7% and 89.0%”

Response: Thank you for the comment. This number has now been added as per your suggestion. Changes have been highlighted in red as the following:

Revised text: 

while in the non-farm economic activities and hired labor reached up to 78.7% and 89.0%, respectively (Line 182). 

17. Line 280: “has been done” – replace with “which has been done”

Response: Thank you for the comment. This phrase has now been replaced as per your suggestion. Changes have been highlighted in red as the following:

Revised text: 

which have been done for all five final empowerment domains and the final EI; (Line 288). 

18. Footnote on Line 283: “weight order” is not clear, perhaps you mean “estimated size of the coefficients”?

Response: Thank you for the comment. This sentence has now been rewriten as per your suggestion. Changes have been highlighted in red as the following:

Revised text: 

However, the estimated size of the factors was different between the two models. (The footnote, page 16)

19. Footnote on Line 283: “impact on empowerment” – replace with: “association with empowerment”

Response: Thank you for the suggestion. This phrase has now been replaced as per your suggestion. Changes have been highlighted in red as the following:

Revised text: 

While in the binary regression model, the education variable had the greatest association with empowerment ( The footnote, page 16).

20. Footnote on Line 283: “rural areas, besides the the policies” – replace with “rural areas, in addition to policies”

Response: Thank you for the suggestion. This phrase has now been replaced as per your suggestion. Changes have been highlighted in red as the following:

Revised text: 

These results indicated that in order to improve the level of women's empowerment in rural areas, in addition to policies ( The footnote, page 16). 

21. Footnote on Line 283: I suggest adding this final sentence in the footnote: “These results are not presented in the paper, but are available upon request.”

Response: Thank you for the suggestion. This sentence has now been added as per your suggestion. Changes have been highlighted in red as the following:

Revised text: 

These results are not presented in the paper, but are available upon request ( The footnote, page 16). 

22. Line 318: “confucius’ views” – should this be capitalized “Confucius’ views” or perhaps replaced with “Confucian views”?

Response: Thank you for the suggestion. This phrase has now been replaced as per your suggestion. Changes have been highlighted in red as the following:

Revised text: 

One of the reasons for this was that the rural society of Vietnam is still being influenced by Confucian views, (Line 326).

23. Line 322: “less rights than men of property ownership” – replace with “less rights than men regarding property ownership” OR “less rights than men to own property”

Response: Thank you for the suggestion. This phrase has now been replaced as per your suggestion. Changes have been highlighted in red as the following:

Revised text: 

In addition, our results also indicated that most of the women in the rural area have less rights than men regarding property ownership (Lines 329 – 330).

24. Line 339: I’m not sure that “endurance” is the right word here. Suggest rephrasing or deleting “endurance and”

Response: Thank you for the suggestion. This word has now been replaced. Changes have been highlighted in red as the following:

Revised text: 

The women in central Vietnam are known for their forbearance and undemanding behavior (Lines 347 – 348). 

25. Line 348: “The time use is” – replace with “Time use is”

Response: Thank you for the suggestion. This phrase has now been replaced as per your suggestion. Changes have been highlighted in red as the following:

Revised text: 

Time use is an indicator that was clearly different between men and women (Line 356). 

26. Lines 353-354: “considered it as a women’s duty” – replace with: “considered it as the women’s duty”

Response: Thank you for the suggestion. This phrase has now been replaced as per your suggestion. Changes have been highlighted in red as the following:

Revised text: 

They often shun the housework and considered it as the women's duty (Lines 361 - 362).

27. Line 355: “women have much lesser rights” – replace with: “women have fewer rights”

Response: Thank you for the suggestion. This phrase has now been replaced as per your suggestion. Changes have been highlighted in red as the following:

Revised text: 

Taken together, our results showed that women have fewer rights than men in all 5 domains (Line 363). 

28. Line 365: “had lower voices” – replace with: “had less voice”

Response: Thank you for the suggestion. This phrase has now been replaced as per your suggestion. Changes have been highlighted in red as the following:

Revised text: 

This fact has been one of the main reasons why women in this study had less voice (Line 373). 

29. Line 377: “typical elements” – replace with: “typical contributors to women’s disempowerment”

Response: Thank you for the suggestion. This phrase has now been replaced as per your suggestion. Changes have been highlighted in red as the following:

Revised text: 

Resource ownership, spending time, and household financial control were typical contributors to women’s disempowerment (Lines 385 – 386). 

30. Line 392: “it is required” – replace with “it requires”

Response: Thank you for the suggestion. This phrase has now been replaced as per your suggestion. Changes have been highlighted in red as the following:

Revised text: 

" To be able to escape from this vicious circle it requires a "kick from outside" (Line 401)

Thank you very much for all your valuable comments and suggestions!

28 May 2023

To

Reviewer #3

Dear Reviewer,

We would like to thank the reviewer for the insightful review and valuable comments on our manuscript. In the revised version, we hope that we have satisfactorily addressed the comments raised by the reviewer. 

Here below, we provide a point-to-point answer to all remarks. All the changes have been highlighted in the manuscript in red and a clean version with line numbers was also provided.

We would like to thank the reviewer again for taking the time to review our manuscript. We hope the manuscript after careful revisions meet your high standards. We are happy to respond to any further questions and comments you may have.

Yours sincerely,

Thanh Tung Le

Reviewer #3

1. The title does not fully reflect the content of the study. The title should reflect gender differences in relation to agricultural land tenure and decisions about agricultural activities. That's because in all sections, including results, discussions and conclusions, there are clearly presented differences between men and women related to these activities. I therefore suggest that the abstract part also be rewritten accordingly.

Response: Thank you very much for the suggestion. Our research focuses on evaluating gender differences. More specifically, the gender disparity in the index of empowerment, where agricultural land use rights and decision-making in agricultural activities are two out of the ten factors that constitute the overall empowerment level.

Because of this reason, our originally planned title was "Gender Gap in Empowerment in Agriculture among ethnic minorities in Central Vietnam." However, after completing the manuscript, we realized that the majority of women were not empowered. Therefore, we wanted to adjust the title of the study to focus more on women's empowerment. That's the reason, in the first manuscript submitted to the journal, our research was titled "Level of Women' empowerment in Agriculture of Ethnic Minorities of Central Vietnam and Influencing Factors". 

After two rounds of review, both reviewers, especially the first reviewer, suggested that we consider adjusting the title of the study to "Ethnic minority women’s Empowerment in Agriculture in the central region of Viet Nam".

We realized this to be the most accurate title that reflects the aim of this research. Therefore, We would like to keep the title of the study as it is in the manuscript.

Following the reviewer's suggestion, the study's abstract was adjusted to clarify the gender gap in empowerment further. Changes have been highlighted in red as the following:

Revised text: 

Gender inequality and women's empowerment are two closely related issues. While the gender inequality index has been assessed by different studies, that of women's empowerment remained limited. In the present work, we attempted to evaluate the women's empowerment index by comparing it with the male partner's empowerment index in the same household. We used the Women’s Empowerment in Agriculture Index (WEAI) as a framework for reference. A questionnaire was designed to interview 300 people including both men and women in the same ethnic minority household in central Vietnam. The difference in the empowerment level between men and women was assessed through five-component empowerment indicators: agricultural participation, resource ownership, financial control, social organizations participation, and time usage. The results showed that up to 70% of women were disempowered compared to only 15% of men. The binary logistic model revealed the age at first marriage, the level of children's education, education level, distance to the nearest urban area, and the number of children were associated with women's empowerment; whereas age, income, and the level of gender awareness did not show any correlation. (Lines 29 – 42). 

2. Introduction should be provided with information about Vietnam's social context on women's and men's empowerment in terms of property ownership, power between men and women in social life and employment, especially in terms of agricultural land use. In addition, policies on agricultural land use rights are provided in more detail so that readers can better understand this field in Vietnam. From there, give the significance of the research when practiced in Vietnam.

Response: Thank you for the suggestion. The introduction of manuscript has now been changed as per your suggestion. Changes have been highlighted in red as the following:

Revised text: 

In Vietnam, research on empowerment is often associated with research on gender equality. Therefore, There have been very few separate studies on gender empowerment in Vietnam to date. An annual national study on gender inequality mentioned that the rights of women in economics and politics showed that women often have less opportunity to make decisions in economics and politics than men. Even, in many cases, they had no right to make any decisions in their family life [26]. The main reasons for this could be due to cultural barriers, low educational attainment, and lack of opportunities to own productive resources such as land, capital, and property [26]. Noticeably, there are no studies in Vietnam that have measured the overall index of women's empowerment in agriculture to date (Lines 75 – 83). 

3. Research data: Authors need to supplement information on economic, social, cultural characteristics, agricultural land use rates, occupational patterns in the localities to collect data to help readers understand why research to decide on the selection of information collection in these localities.

Response: Thank you for the suggestion. The research data of manuscript has now been changed as per your suggestion. Changes have been highlighted in red as the following:

Revised text: 

Like most other ethnic minorities, the majority of ethnic minorities in Central Vietnam live in special areas such as mountainous regions, border areas, and regions prone to frequent natural disasters, consequently, their life is always faced with many difficulties. In this study, we selected three areas with the highest percentage of ethnic minorities in the Central region to get research samples. These are areas with a very high poverty rate, low educational attainment, and many unsound customs such as child marriage and closely consanguineous marriages (Lines 90 – 96). 

4. Data analysis: I think the results of the logistic regression model are not strong enough. Because factors from the husband's side can play very important role related to women' empowerment in the agricultural sector, such as husband's education, age gap between husband and wife, and husband's occupation. In Vietnamese culture, women are required to be subordient for 3 men in their life, including father's subordient when unmarried, husband's subordient when married and son's subordient when husband died. Therefore, the regression model is flawed when it does not take into account the husband's characteristics in women's empowerment.

Response: thank you very much for your valuable question. To answer the question, we would like to provide the following response:

Firstly, during the ideation and discussion process to conduct this research, the authors in our research team were well aware of the factors from the husband's side that could influence the level of empowerment of the wife. Therefore, in the initial pilot questionnaire, we included several factors such as occupation, habits, and the extent of the husband's involvement in non-agricultural activities. However, after completing the first round of the pilot survey, we noticed that in the particular minority ethnic area where we conducted the study, most of the characteristics we included in the questionnaire were highly homogeneous across almost all respondents. For example, 100% of the husbands were engaged in agricultural work, and even 100% of the wives were also involved in agricultural work and household chores. Moreover, this was a very long questionnaire that had to be administered at nearly the same time (consecutively) to both the wife and the husband within the same household. Therefore, to simplify the questionnaire, we decided to eliminate all non-statistically significant factors, including the occupation of both the wife and the husband.

Secondly, the remaining two factors related to the husband's characteristics in our final version of the questionnaire were age and education level. Prior to data analysis, we expected the husband's education level to be a significant variable. However, when analyzed using regression models (both multiple regression and binary logistic regression models), neither of these two factors had any influence on the level of women's empowerment.

Thirdly, as mentioned in previous responses, this is a study on empowerment in a specific minority ethnic area of Vietnam. Therefore, we want to focus on the main factors that contribute to suggesting appropriate policies. That is why our regression model only includes variables as presented in the manuscript.

Finally, it is possible that these findings are specific to a particular minority ethnic area, such as the one we studied. In another recent study, we conducted in a different non-minority ethnic area, the education level of the husband had a clear impact on the level of women's empowerment.

5. Policy recommendations: As I mentioned in the analysis of the results, because the research is lacking in taking into account the husband's characteristics in women's empowerment. Therefore, suggesting policies of this study may be flawed if only focusing on policies for women, while Vietnamese women are still heavily influenced by their husbands and in-laws.

Response: thank you very much for your comment. As mentioned in our response to question 4, our focus is solely on providing policy recommendations based on the outcomes of the regression model we conducted. We intend to incorporate additional policy recommendations in an other study, as we identify further factors that influence the level of women's empowerment.

Thank you very much for all your valuable comments and suggestions!

---

## [Editor Report · Decision Letter 3]

31 May 2023

Ethnic minority women’s empowerment in agriculture in the central region of Viet Nam

PONE-D-21-33301R3

Dear Dr. LE,

We’re pleased to inform you that your manuscript has been judged scientifically suitable for publication and will be formally accepted for publication once it meets all outstanding technical requirements.

Kind regards,

Nai-peng Tey, Ph.D

Guest Editor

PLOS ONE

Additional Editor Comments (optional):

Dear authors,

I thank you for your efforts in making changes to your manuscript, based on the comments of the three reviewers. I suggest you include some of your responses to reviewer 3's comments and suggestions before you finalize the paper.

Best regards.

Nai Peng TEY
---

## [Editor Report · Acceptance letter]

28 Jul 2023

PONE-D-21-33301R3 

Ethnic minority women’s empowerment in agriculture in the
central region of Viet Nam 

Dear Dr. Le:

I'm pleased to inform you that your manuscript has been deemed suitable for publication in PLOS ONE. Congratulations! Your manuscript is now with our production department. 

Kind regards, 

on behalf of

Dr. Nai-peng Tey 

%CORR_ED_EDITOR_ROLE%

PLOS ONE